# A hollow-tube-like hydrospongel for multimodal therapy of advanced colorectal cancer

Tao Wu [1,2,7], Tenghui Li[3,4,7], Chengzhi Zhang[3,4], Yu Tian[1], Hao Li[1], Yuxin He[5], Xu Yan[6], Tianxing Gong [1] ✉, Junhua Zhao[3,4] ✉ & Zhenning Wang [3,4] ✉

Preserving anal function in advanced-stage colorectal cancer (CRC) treatment remains a significant challenge. Here we introduce a hollow-tube-like hydrospongel (HTHSG) as an advanced neoadjuvant therapy. Constructed from cellulose nanofibers crosslinked with $Fe_3O_4$@PDA (polydopamine) nanoparticles, the HTHSG combines quick swelling (~10 seconds) with high fracture strength (>250 kPa) for enhanced mechanical stability. The hydrospongel enables precise, localized delivery of the chemotherapeutic agent 5-FU directly to the tumor site. Beyond conventional chemotherapy, HTHSG employs electromagnetic induction to achieve targeted thermal ablation and chemodynamic therapy, minimizing collateral damage to healthy tissues. Comparative studies in orthotopic, cell-derived, and patient-derived xenograft models demonstrate the superior tumor-reduction efficacy of HTHSG over traditional neoadjuvant therapies. Feasibility studies in a beagle model and human-sized dummy model further validate the HTHSG's potential for clinical application, showing preserved anal function and biocompatibility. These findings establish HTHSG as a promising pre-surgical treatment option for advanced-stage CRC, offering improved therapeutic outcomes and quality of life for patients.

Colorectal cancer (CRC) is one of the most prevalent gastrointestinal malignancies and the third leading cause of cancer-related mortality worldwide, with over 1.85 million new cases and approximately 850,000 deaths reported annually[1,2]. Surgical resection remains the cornerstone of curative treatment[3]. However, for tumors located near the anus, achieving optimal oncologic control while preserving anal function poses a significant clinical challenge. The conventional approach, abdominoperineal resection (APR), effectively eliminates the tumor but necessitates a permanent colostomy, profoundly affecting patients' quality of life due to urogenital dysfunction and psychosocial burden[4,5]. Conversely, local excision preserves the anus but risks incomplete tumor removal due to potential margin involvement[6]. Thus, preoperative strategies aimed at reducing tumor burden and downstaging the disease are essential for improving surgical outcomes and functional preservation.

The standard neoadjuvant chemoradiotherapy (nCRT) regimen aims to downstage tumors before surgery, yet its clinical efficacy is often limited, with some patients exhibiting minimal or no response[7,8]. To overcome these limitations, biomaterial-based drug delivery systems (DDSs) have emerged as promising alternatives, enabling localized drug administration to maximize intratumoral drug concentrations while reducing systemic toxicity[9,10]. Among these,

[1]Department of Biomedical Engineering, Shenyang University of Technology, Shenyang, China. [2]College of Medicine and Biological Information Engineering, Northeastern University, Shenyang, China. [3]Department of Surgical Oncology and General Surgery, The First Hospital of China Medical University, Shenyang, China. [4]Key Laboratory of Precision Diagnosis and Treatment of Gastrointestinal Tumors, Ministry of Education, China Medical University, Shenyang, China. [5]Department of Critical Care Medicine, The First Hospital of China Medical University, Shenyang, China. [6]The VIP Department, School and Hospital of Stomatology, China Medical University, Liaoning Provincial Key Laboratory of Oral Diseases, Shenyang, China. [7]These authors contributed equally: Tao Wu, Tenghui Li. ✉e-mail: tianx.gong@gmail.com; jhzhao@cmu.edu.cn; znwang@cmu.edu.cn

nanoparticle- and hydrogel-based platforms have demonstrated enhanced anti-tumor efficacy with fewer adverse effects compared to conventional systemic chemotherapy[9,11,12]. Moreover, integrating chemotherapy with emerging therapeutic modalities, such as hyperthermia and reactive oxygen species (ROS)-mediated cytotoxicity, has shown potential for overcoming drug resistance in CRC[13]. For example, Zhu et al. developed a dual-drug delivery nanoenzyme that synergistically combined chemotherapy with photothermal therapy, leading to substantial tumor suppression in both subcutaneous and orthotopic CRC models[14]. Similarly, Shen et al. engineered a solid lipid nanoparticle system for chemo/magnetothermal therapy, significantly inhibiting primary colon cancer growth[15]. Despite these advancements, the need for delivering sustained, multimodal, synergistic anti-tumor therapy directly to the tumor site without compromising patient safety remains critical.

In this work, we present a preoperative therapeutic strategy utilizing a hollow-tube-like hydrospongel (HTHSG), which integrates magnetothermal, chemo-, and chemodynamic therapies to enhance tumor regression in colorectal cancer (Fig. 1). Unlike conventional hydrogel-based systems, HTHSG uniquely combines the quick-swelling properties of a sponge with the high mechanical strength of a hydrogel, ensuring robust structural integrity within the colorectal lumen[16]. Engineered from carboxylated cellulose nanofibers (CNF-C) crosslinked with $Fe_3O_4$@PDA nanoparticles (NPs), HTHSG serves as a localized 5-fluorouracil (5-FU) delivery platform with controlled drug release under simulated bowel movement conditions. Upon electromagnetic induction, $Fe_3O_4$ NPs generate heat for tumor ablation and modulate reactive oxygen species (ROS) production, activating chemodynamic therapy (CDT) while preventing excessive iron toxicity[17–19]. Using orthotopic, cell-derived xenograft (CDX), and patient-derived xenograft (PDX) models, we demonstrate HTHSG's superior tumor

suppression. Furthermore, feasibility studies in a large-animal (beagle) model and human-sized dummy model validate its clinical potential, highlighting HTHSG as a highly promising neoadjuvant approach for CRC, particularly for tumors in anatomically challenging locations.

## Results and Discussion

### Synthesis and characterization of $Fe_3O_4$ and $Fe_3O_4$@PDA NPs

The synthesis pathway of $Fe_3O_4$@PDA nanoparticles (NPs) is illustrated in Fig. 2a. $Fe_3O_4$ NPs were prepared using a co-precipitation method, followed by the deposition of a polydopamine (PDA) layer via dopamine self-polymerization in a weakly alkaline solution[20]. The PDA-coated $Fe_3O_4$ solution exhibited a distinct color change (Fig. 2b) and enhanced near-infrared absorption (Supplementary Fig. 1), confirming successful coating[21]. Transmission electron microscopy (TEM) images showed spherical $Fe_3O_4$ NPs and the presence of a PDA layer in $Fe_3O_4$@PDA NPs (Fig. 2c, d). Dynamic light scattering analysis revealed hydrodynamic diameters of $223.7 \pm 49.4$ nm for $Fe_3O_4$ and $349.7 \pm 82.0$ nm for $Fe_3O_4$@PDA, with the size increase attributed to aggregation in solution (Fig. 2e)[22]. The zeta potential shifted from $1.6 \pm 0.2$ mV for $Fe_3O_4$ to $−23.3 \pm 0.4$ mV for $Fe_3O_4$@PDA due to the negative hydroxyl groups from PDA's catechol units (Fig. 2f)[23]. Fourier transform infrared (FTIR) analysis further confirmed PDA deposition with characteristic C−O and N−H bands at $1279$ cm$^{-1}$ and $1506$ cm$^{-1}$, respectively (Supplementary Fig. 2)[23].

The X-ray diffraction (XRD) analysis was performed to characterize the crystalline structures of the synthesized NPs. As shown in Fig. 2g, a series of characteristic diffraction peaks of $Fe_3O_4$ was consistent with the standard pattern for magnetic $Fe_3O_4$ (JCPDS card no. 19-0629) with a cubic reverse spinel structure[24]. Meanwhile, $Fe_3O_4$@PDA had the same crystalline phase as the pristine $Fe_3O_4$ without additional diffraction peaks, indicating that the modification

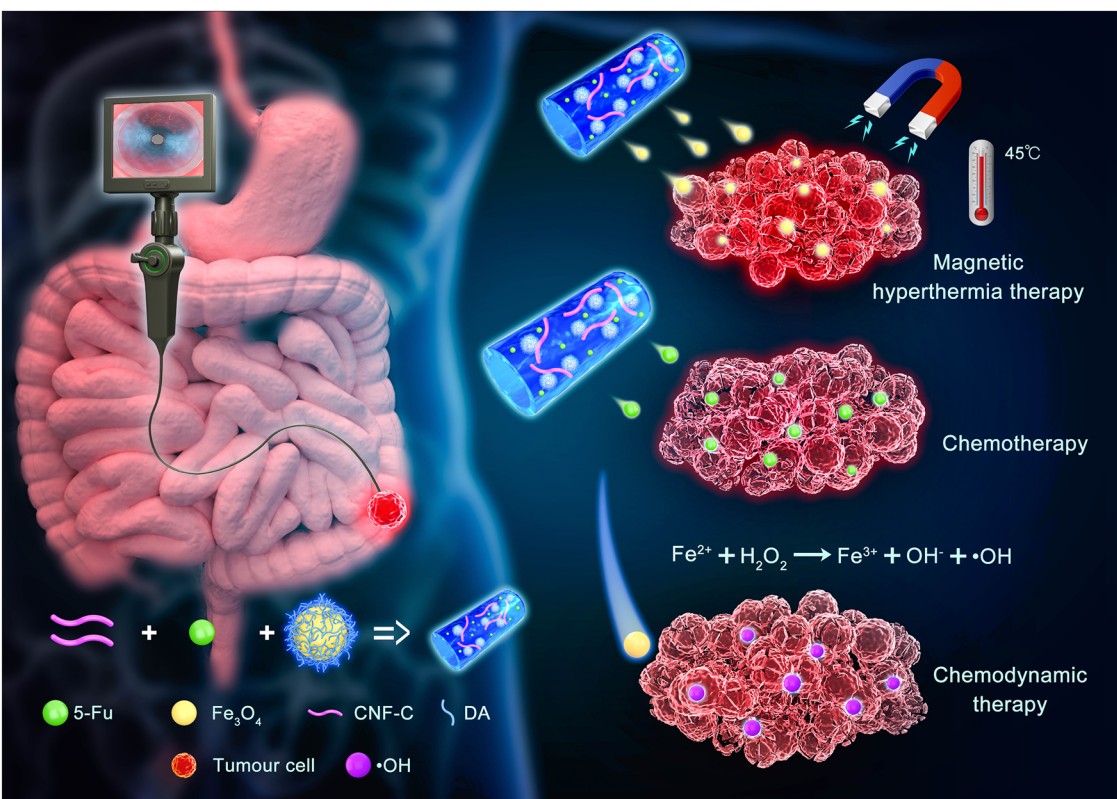

**Fig. 1 | Schematic illustration of HTHSG for treating CRC by magnetothermal-chemo-chemodynamic therapy.** The HTHSG, constructed by crosslinking carboxylated cellulose nanofibers (CNF-C) with $Fe_3O_4$@PDA nanoparticles, functions as a site-specific drug delivery platform enabling sustained release of 5-fluorouracil

(5-FU). By integrating magnetic hyperthermia therapy, chemotherapy, and chemodynamic therapy into a multimodal therapeutic modality, HTHSG holds significant potential as a preoperative therapeutic strategy for advanced colorectal cancer.

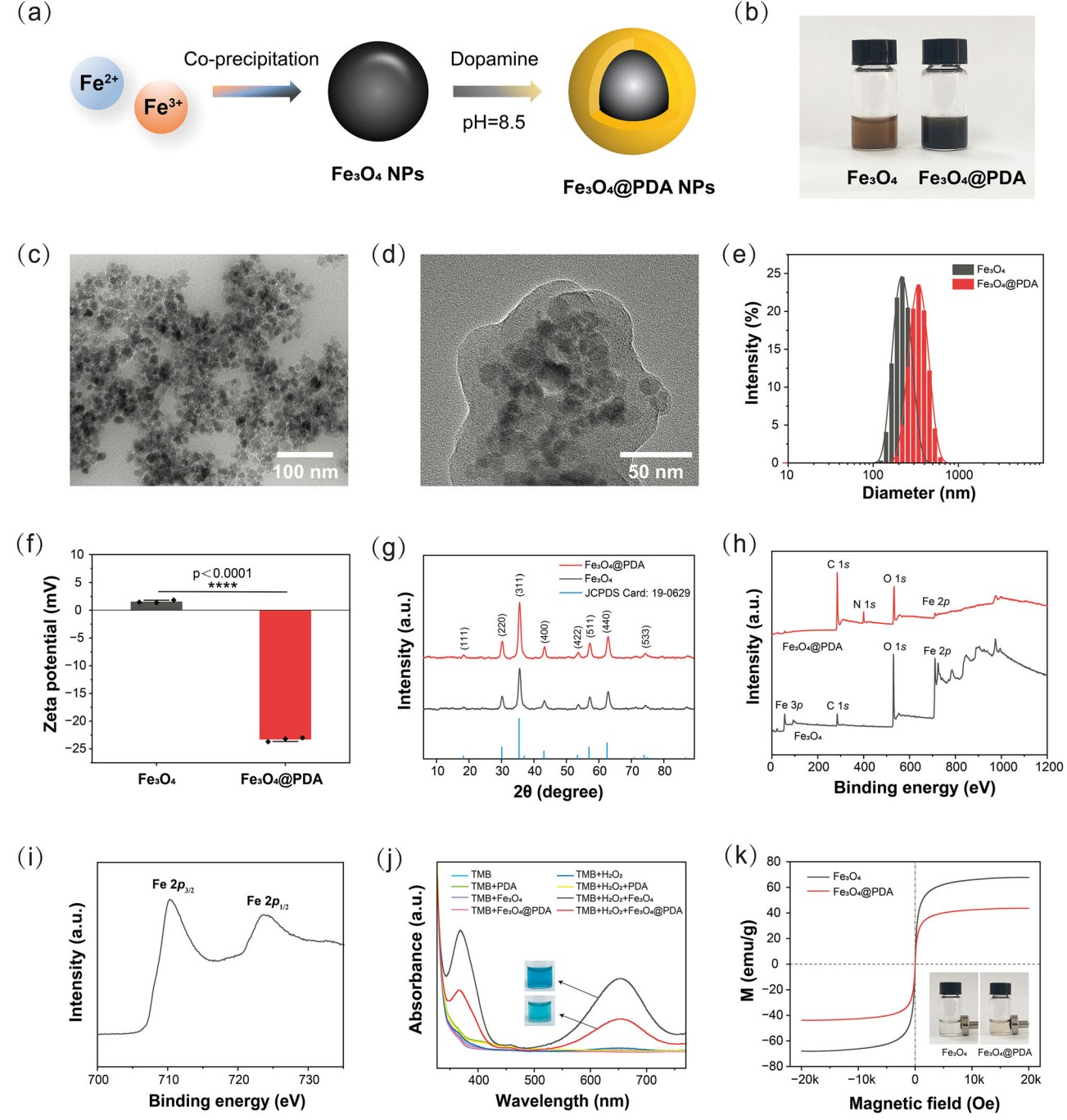

**Fig. 2 | Synthesis and characterization of Fe₃O₄ and Fe₃O₄@PDA NPs.**
**a** Schematic illustration of the synthesis pathway of Fe₃O₄@PDA NPs. **b** Gross appearance of Fe₃O₄ and Fe₃O₄@PDA NPs in aqueous solution. **c**, TEM images of Fe₃O₄ and **d** Fe₃O₄@PDA NPs. **e** Hydrodynamic diameters and **f** Zeta potentials of Fe₃O₄ and Fe₃O₄@PDA NPs. The data were presented as the mean ± SD. ($n = 3$ independent samples). Statistical differences were analyzed by two-tailed unpaired Student's $t$-test. (****$p < 0.0001$). **g** XRD patterns of Fe₃O₄ NPs, Fe₃O₄@PDA NPs, and JCPDS Card: 19-0629. **h** XPS spectra of Fe₃O₄ and Fe₃O₄@PDA NPs. **i**, XPS spectra of Fe $2p$ in Fe₃O₄ NPs. **j** UV-Vis spectra and photographic images of PDA, Fe₃O₄, and Fe₃O₄@PDA NPs in TMB solution with and without H₂O₂. **k** Magnetic hysteresis loops of Fe₃O₄ and Fe₃O₄@PDA NPs. Source data are provided as a Source Data file.

with PDA did not alter the crystalline structure of Fe₃O₄ NPs. Since the XRD diffraction pattern of γ-Fe₂O₃ is similar to Fe₃O₄[25], we further performed the X-ray photoelectron spectra (XPS) to characterize the elemental composition of the NPs; the survey spectra are shown in Fig. 2h. Fe and O were predominantly present in Fe₃O₄, while little C can be attributed to slight external carbon contamination[26]. In comparison, extra N was present in Fe₃O₄@PDA, attesting to the successful coating with PDA. The peaks with binding energies of 710.4 and 724.0 eV corresponded to Fe $2p_{3/2}$ and Fe $2p_{1/2}$, respectively (Fig. 2i),

and the absence of the characteristic satellite peak of γ-Fe₂O₃ at 719.2 eV indicates that the particles are pure Fe₃O₄[27].

The tumor microenvironment (TME) is characterized by weak acidity and high endogenous H₂O₂ levels (100 μM−1 mM), and Fe²⁺ is capable of generating highly toxic •OH from H₂O₂ via the Fenton reaction to kill tumor cells[28]. To test whether the Fenton reaction was occurring with NPs, 3,3′,5,5′-tetramethylbenzidine dihydrochloride hydrate (TMB) was chosen as an indicator for •OH generation, as it can be oxidized into blue oxTMB by •OH and possessed a characteristic

absorption peak at 652 nm[29]. As depicted in Fig. 2j, only the TMB solution containing both $H_2O_2$ and NPs showed a visible color change and presence of the characteristic absorption peaks upon 1 h of incubation. This suggests that the prepared NPs have the potential to release $Fe^{2+}$ and generate •OH with intracellular $H_2O_2$ in the TME, thus promoting tumor cell apoptosis. In addition, the •OH production in $Fe_3O_4$@PDA was lower than in $Fe_3O_4$ because the PDA shells are degraded first in a weakly acidic environment (pH = 6.0)[30], consuming the $H^+$ and slowing down $Fe^{2+}$ release.

Finally, magnetic hysteresis loops showed saturation magnetizations of 67.8 emu/g for $Fe_3O_4$ and 43.6 emu/g for $Fe_3O_4$@PDA, with reduced magnetization attributed to the diamagnetic PDA layer (Fig. 2k)[31]. Both NPs exhibited superparamagnetic behavior, as evidenced by the lack of remanence and coercivity[24].

### Synthesis and characterization of hollow-tube-like hydrospongel (HTHSG)

Hydrospongels were fabricated by covalently crosslinking carboxylated cellulose nanofiber (CNF-C) with $Fe_3O_4$@PDA nanoparticles (NPs). Three hydrospongels were prepared with varying $Fe_3O_4$@PDA NPs concentrations (10, 20, and 30 mg/mL), termed CFP10, CFP20, and CFP30, respectively. A control hydrospongel with 10 mg/mL of PDA NPs was designated CP10. CNF-C was covalently crosslinked with synthesized $Fe_3O_4$@PDA NPs to fabricate hydrospongels.

**Swelling and mechanical properties.** The hydrospongels exhibited quick swelling, achieving ~95% equilibrium within 10 seconds and full equilibrium by 60 minutes. Swelling ratios for CP10, CFP10, CFP20, and CFP30 were 1659%, 1610%, 1446%, and 1305%, respectively (Fig. 3a; Supplementary Fig. 3). To avoid the effects of the freeze-drying process on the hydrospongel volume, the swelling ratio was calculated from the mass change of the lyophilized hydrospongel before and after immersion in PBS solution. Compressive tests showed that CP10 had higher fracture strength and toughness than CFP10, attributed to its higher PDA content. Both parameters increased with $Fe_3O_4$@PDA NPs concentration in CFP20 and CFP30, while compressive modulus remained consistent across all formulations (Fig. 3b, c; Supplementary Fig. 4).

**Magnetic hyperthermia and biocompatibility.** $Fe_3O_4$ NPs generate heat under alternating magnetic fields (AMF) via Neel and Brownian relaxations[32], capable of inducing tumor cell death at temperatures above 42 °C[33,34]. Magnetic hyperthermia tests revealed temperature increases proportional to $Fe_3O_4$@PDA NPs content: 13.9 °C (CFP10), 18.6 °C (CFP20), and 23.2 °C (CFP30) after 10 minutes of AMF exposure (Fig. 3d, e). CFP20 (45.7 °C) was selected for further experiments as it aligns with thermal ablation thresholds while avoiding excessive heating[35]. Biocompatibility tests showed no adverse effects on normal intestinal cells (NCM 460) after 3 days of culture (Supplementary Fig. 5).

**Structural and functional characterization.** HTHSG was cast using a 3D printed mold (Supplementary Fig. 6). FTIR spectra confirmed covalent crosslinking between CNF-C and PDA (Supplementary Fig. 7), while EDS analysis showed uniform distribution of C, N, O, and Fe in CFP20, indicating successful integration of $Fe_3O_4$@PDA NPs (Fig. 3f, g; Supplementary Fig. 8)[36].

CFP20 hydrospongels produced hydroxyl radicals (•OH) via the Fenton reaction, with higher •OH production observed at 45 °C than 37 °C (Fig. 3h). The release of iron ions from HTHSG was further investigated to determine the mechanism for initiating the Fenton reaction. As shown in Supplementary Fig. 9, a larger amount of iron ions was released from HTHSG after AMF treatment ($p < 0.0001$), thereby facilitating the Fenton reaction.

Structural integrity tests under cyclic compression demonstrated resilience, with no rupture after 50 cycles, and stability under simulated bowel movements indicated position retention and durability

(Supplementary Fig. 10; Supplementary Movie 1). These results indicated that HTHSG provides favorable stability and position retention and that it can be subjected to a low-residual diet.

To avoid excessive release of NPs, and thus attain a long-term magnetothermal effect on tumor cells, the $Fe_3O_4$@PDA NPs were covalently crosslinked into the hydrospongel. The degradation of the CFP20 hollow tube is shown in Supplementary Fig. 11. The hollow tube in PBS solution showed no significant weight change within 20 d, indicating minimal degradation of the hollow tube. Due to the presence of various enzymes in the intestine, which may result in enzymatic degradation of hydrospongel, we further examined the degradation of the CFP20 hollow tube when treated with lysozyme, which is widely distributed across human bodily fluids and organs[37,38]. The weight of the hollow tube increased significantly without degradation, resulting from the positively charged lysozyme bonding to the -$COO^-$ in the CNF-C through electrostatic interactions, causing an increased mass[39].

**Drug release and thermal simulation.** The HTHSG can serve as a carrier for chemotherapeutic agents such as 5-FU to drive synergistic anti-tumor efficacy. We dropwised 5-FU solution to CFP20 hydrospongels. HTHSGs loaded with 5-FU exhibited rapid initial release (86.5% in 20 minutes) and slowed release over 1 hour, reaching 92% (Supplementary Figs. 12, 13). To assess the clinical application potential of our hydrospongel, we conducted a thermal simulation analysis using COMSOL Multiphysics. Colorectal cancer usually infiltrates into the intestinal wall and grows in a circular manner along the wall[40]. Tumors are usually a circular or semi-circular shape[40] around 5 cm × 5 cm × 1 cm in size, with 5 cm × 5 cm representing the tumor size along the intestinal wall[41], and 1 cm representing the thickness of the tumor[42]. Therefore, COMSOL simulations modeled colorectal tumors (5 cm × 5 cm × 1 cm) with thermal conductivity values for tumor and intestinal wall set to 0.5 and 0.542 W/m·K[43,44], respectively. When CFP20 was heated to 45 °C, tumor temperatures at the tumor center reached 42 °C within 6 minutes and 43 °C within 9 minutes (Fig. 3i–m). We also built a 5 cm × 5 cm × 1 cm phantom tumor with thermal conductivity of 0.5 W/m·K using Gelma and collagen I. Adhering to CFP20 in AMF for 10 minutes, the opposite side of phantom could reach 40 °C, achieving effective heat penetration across the tumor. (Fig. 3n)

### Anti-tumor effect and biosafety of HTHSG in vitro

To investigate its anti-tumor potential, CFP20 hydrospongel, which demonstrated optimal thermal ablation properties under electromagnetic induction, was selected for in vitro studies. Initial safety evaluations using CP10 (PDA-only hydrospongel) showed no significant cytotoxicity (Fig. 4a–c).

**Magnetothermal therapy and CDT effects.** HCT-116 colorectal cancer cells were divided into CFP20 (no AMF exposure) and CFP20 + AMF (10 minutes AMF exposure) groups. The CFP20 group showed no reduction in cell viability or increased apoptosis compared to the control group, indicating the material's safety without AMF. In contrast, AMF exposure significantly decreased cell viability ($0.126 \pm 0.04$ for CFP20 + AMF vs. $0.503 \pm 0.10$ for the control at 72 h, $p < 0.0001$) and increased apoptotic cells ($16.91 \pm 2.30\%$ for CFP20 + AMF vs. $4.74 \pm 1.74\%$ for the control, $p < 0.0001$; Fig. 4a–c). These results confirm that CFP20 effectively kills tumor cells through magnetothermal therapy. When the appropriate electromagnetic induction is applied, the magnetic moment of $Fe_3O_4$ NPs rotates, generating heat and transferring to surrounding cells and effectively killing cancer cells[45].

To explore the mechanism of inducing cellular apoptosis, intracellular •OH production was analyzed using the DCFH-DA assay. Bright green fluorescence was observed only in the CFP20 + AMF group, indicating that electromagnetic induction triggered $Fe^{2+}$ release, leading to •OH production via the Fenton reaction (Fig. 4d,

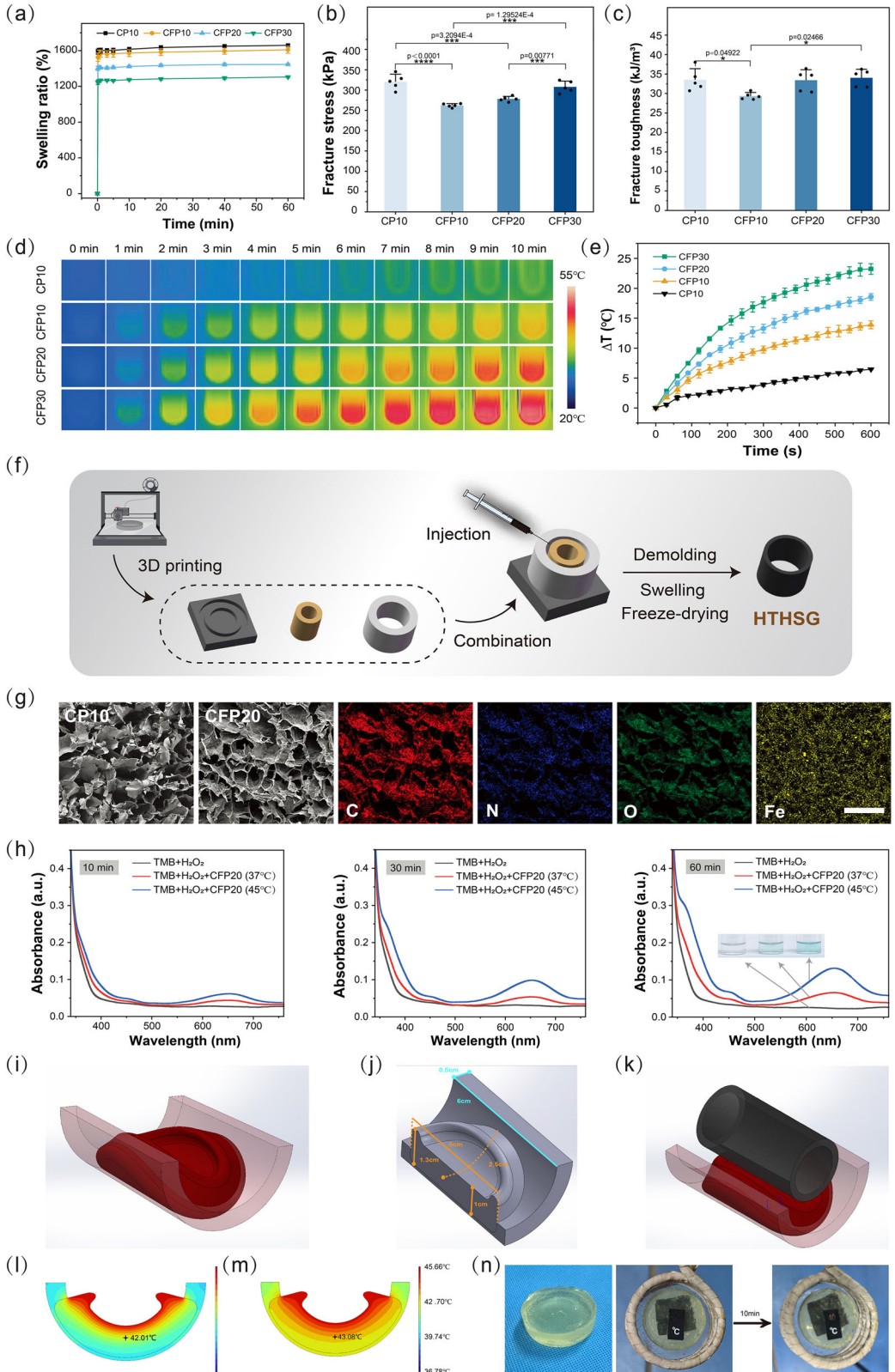

Supplementary Fig. 14). Western blot analysis revealed elevated Cleaved Caspase 3 (apoptotic marker)[46] and heat shock protein 70 (HSP70) levels, along with decreased glutathione peroxidase 4 (GPX4) (Fig. 4e), confirming apoptosis via thermal ablation and glutathione depletion. The glutathione depletion in the CFP20 + AMF group was also confirmed by the UPLC-PDA method[47] (Supplementary Fig. 15). Intracellular HSP70 overexpression is thought to be caused by high-

temperature stimulation[48]. GPX4 downregulation is believed to be related to depleted glutathione (GSH)[49,50]. These results suggest that thermal ablation by CFP20 under AMF also serves as a switch to enhance CDT.

**Synergistic effects of drug-loaded HTHSG.** To evaluate the combined anti-tumor effect, HCT-116 cells were treated with 5-FU,

**Fig. 3 | Synthesis and characterization of HTHSG. a** Swelling behavior of hydrospongels in 60 minutes. The data were presented as the mean ± SD ($n = 5$ independent samples). **b** The fracture stress and **c**, fracture toughness of the swollen hydrospongels. The data were presented as the mean ± SD ($n = 5$ independent samples). Statistical differences were analyzed by one-way ANOVA first, and then by Tukey's post hoc test. (*$p < 0.05$, ***$p < 0.001$ and ****$p < 0.0001$). **d** Thermal images and **e**, temperature profiles of swollen hydrospongels with different nanoparticle contents under electromagnetic induction for 10 minutes. The data were presented as the mean ± SD ($n = 3$ independent samples). **f** The schematic diagram of the preparation of HTHSG, the 3D-printed mold was modeled using SolidWorks 2023. **g** The SEM images of CP10 and CFP20, and the elemental mappings of C, N, O, and Fe in CFP20. Scale bar = 500 μm. **h** UV-Vis spectra and photographic images of CFP20 incubated at 37 °C and 45 °C in TMB solution containing $H_2O_2$. **i** Modeling of

colorectal tumor (dark red) on the intestinal wall (light red) using COMSOL Multiphysics 6.2 software. **j** Longitudinal sectional view along the circular axis of the colorectal cancer model with dimensions shown, modeling using COMSOL Multiphysics 6.2 software. **k** Modeling hydrospongel (black) for colorectal cancer thermal simulation analysis using COMSOL Multiphysics 6.2 software. **l** The temperature distribution after 6 minutes of heating. Sectional view perpendicular to the circular axis of the colorectal tumor model. **m** Temperature distribution after 9 minutes of heating. Sectional view perpendicular to the circular axis of the colorectal tumor model. **n** The Phantom tumor adhered to CFP20 in AMF for 10 minutes, the opposite side of the phantom could reach 40 °C, as measured by the surface temperature indicating strip (when the temperature reaches 40 °C, the number 40 in the strip appeared). Source data are provided as a Source Data file.

CFP20 + AMF, and CFP20 + 5-FU + AMF. Fluorouracil-based chemotherapy regimen is currently the clinical first-line chemotherapy for CRC patients[51]. Therefore, 5-FU was selected as an active control. The CFP20 + 5-FU + AMF group exhibited the most pronounced reduction in cell viability (0.06 ± 0.02 vs. 0.16 ± 0.02 for 5-FU at 72 h, $p < 0.0001$) and higher apoptosis rates (24.65 ± 2.14% vs. 15.25 ± 1.06% for 5-FU, $p < 0.01$; Fig. 4f, g, Supplementary Fig. 16). Western blot analysis (Supplementary Fig. 17) showed elevated Cleaved Caspase 3 in the 5-FU, CFP20 + AMF, and CFP20 + 5-FU + AMF groups, with the highest levels in the CFP20 + 5-FU + AMF group. Increased HSP70 and decreased GPX4 were observed only in CFP20 + AMF and CFP20 + 5-FU + AMF groups, indicating the additional effects of thermal ablation and CDT in the latter. 5-FU is known to induce apoptosis of tumor cells accompanied by cleavage activation of Caspase 3[52,53]. The increased HSP70 and the decreased GPX4 levels confirm that the CFP20 and CFP20 + 5-FU hydrospongel exerts hyperthermia and the Fenton reaction under electromagnetic induction. These results demonstrate that CFP20 hydrospongel synergistically enhances traditional chemotherapy by integrating thermal ablation, CDT, and 5-FU delivery, offering superior anti-tumor efficacy compared to individual treatments.

## Anti-tumor effect of HTHSG in cell-derived xenograft (CDX) Model

To evaluate the in vivo anti-tumor efficacy of HTHSG, we constructed a cell-derived xenograft (CDX) model using HCT-116 colorectal cancer cells. To prolong the release of 5-FU, a small hydrophilic chemotherapeutic drug, we fabricated drug-loaded hydrogel microspheres using microfluidic technology (Fig. 4h and Supplementary Fig. 18). These microspheres, composed of gelatin and dialdehyde cellulose (DAC), were crosslinked via Schiff base reactions, as confirmed by the imine bonds observed at 1674 cm$^{-1}$ in FTIR spectra[54] (Fig. 4i). Microscopic imaging revealed that the droplets produced by the microfluidic device were spherical with an average diameter of 93.17 ± 9.84 μm (Fig. 4j, k) and remained intact after crosslinking and cleaning (Fig. 4l). The hydrogel microspheres were then incorporated into HTHSG, resulting in a sustained release of 5-FU over 48 hours in PBS containing 20% serum (Fig. 4m). Within the first 2 hours, 63.37 ± 3.41% of 5-FU was released, and by 48 hours, the release reached 97.71 ± 1.15%.

The CDX model was established by subcutaneously implanting HCT-116 cells into the flanks of mice, followed by treatment with different therapeutic interventions (Fig. 5a). A total of thirty-five tumor-bearing mice were divided into seven groups to compare the anti-tumor effects of HTHSG against conventional chemotherapy and magnetothermal therapy. These groups included Group 1 (PBS + AMF group, intraperitoneally injected with 100 μL PBS and exposed to AMF), Group 2 (CP10 + AMF group, implanted subcutaneously with 25 mg CP10 hydrospongel around the tumor and exposed to AMF), Group 3 (CFP20 group, implanted subcutaneously with 25 mg CFP20 hydrospongel around the tumor), Group 4 (CP20 + 5-FU MSs+AMF group, implanted subcutaneously with 25 mg CP20 + 5-FU MSs

hydrospongel around the tumor and exposed to AMF), Group 5 (5-FU + AMF group, intraperitoneally injected with 100 μL 5-FU solution and exposed to AMF), Group 6 (CFP20 + AMF group, implanted subcutaneously with 25 mg CFP20 hydrospongel around the tumor and exposed to AMF), and Group 7 (CFP20 + 5-FU MSs+AMF group, implanted subcutaneously with 25 mg CFP20 + 5-FU MSs hydrospongel around the tumor and exposed to AMF). Intraperitoneal injections of 5-FU were used to simulate systemic chemotherapy in clinical colorectal cancer (CRC) treatments, as intraperitoneal injection is a classic systemic administration method with little harm and high bioavailability in animal experiments[55,56]. For the hydrospongel-treated groups, the hydrospongels were implanted subcutaneously around the tumor, ensuring close contact with the tumor surface. AMF exposure was applied for 10 minutes every other day over an 11-day treatment period, and local tumor temperature changes were recorded using infrared thermal imaging.

Under electromagnetic induction, the local tumor temperature in mice treated with CFP20 + AMF and CFP20 + 5-FU MSs+AMF groups rapidly increased to ~45 °C, a temperature conducive to thermal ablation (Fig. 5b). In contrast, minimal temperature changes were observed in the control and other groups. During the administration period, the weight of mice was recorded and no significant variations were observed (Fig. 5c). Tumor growth and weight were monitored throughout the experiment, and on the seventh day after treatment, tumor volume and weight were significantly reduced in groups treated with 5-FU or exposed to AMF (Fig. 5d–f and Supplementary table 1). Further immunohistochemical analysis revealed that groups treated with magnetothermal therapy, 5-FU, or their combination showed reduced expression of the proliferation marker KI67 and increased levels of Cleaved Caspase 3, an apoptotic marker (Fig. 5g). Groups treated with magnetothermal therapy (CFP20 + AMF and CFP20 + 5-FU MSs+AMF) also demonstrated decreased GPX4 expression and increased HSP70 levels, indicating the synergistic action of thermal ablation and CDT. The combination treatment group (CFP20 + 5-FU MSs+AMF) exhibited the most substantial changes in these markers, confirming the enhanced anti-tumor efficacy of integrating magnetothermal therapy, CDT, and chemotherapy. Importantly, no signs of systemic toxicity were observed among the groups, and H&E staining (Supplementary Fig. 19) of major organs showed no evidence of organ damage, highlighting the biocompatibility of the HTHSG treatments.

These findings underscore the potential of HTHSG as an innovative preoperative therapeutic strategy for CRC. By combining thermal ablation, chemotherapy, and CDT, HTHSG achieves localized tumor control, significantly reducing tumor size. This approach enhances the likelihood of radical tumor resection and anal preservation, offering a promising alternative for the preoperative management of distal CRC.

## Anti-tumor effect of HTHSG in a patient-derived xenograft (PDX) Model

To further evaluate the potential of HTHSG as an effective preoperative treatment for colorectal cancer (CRC), we constructed a

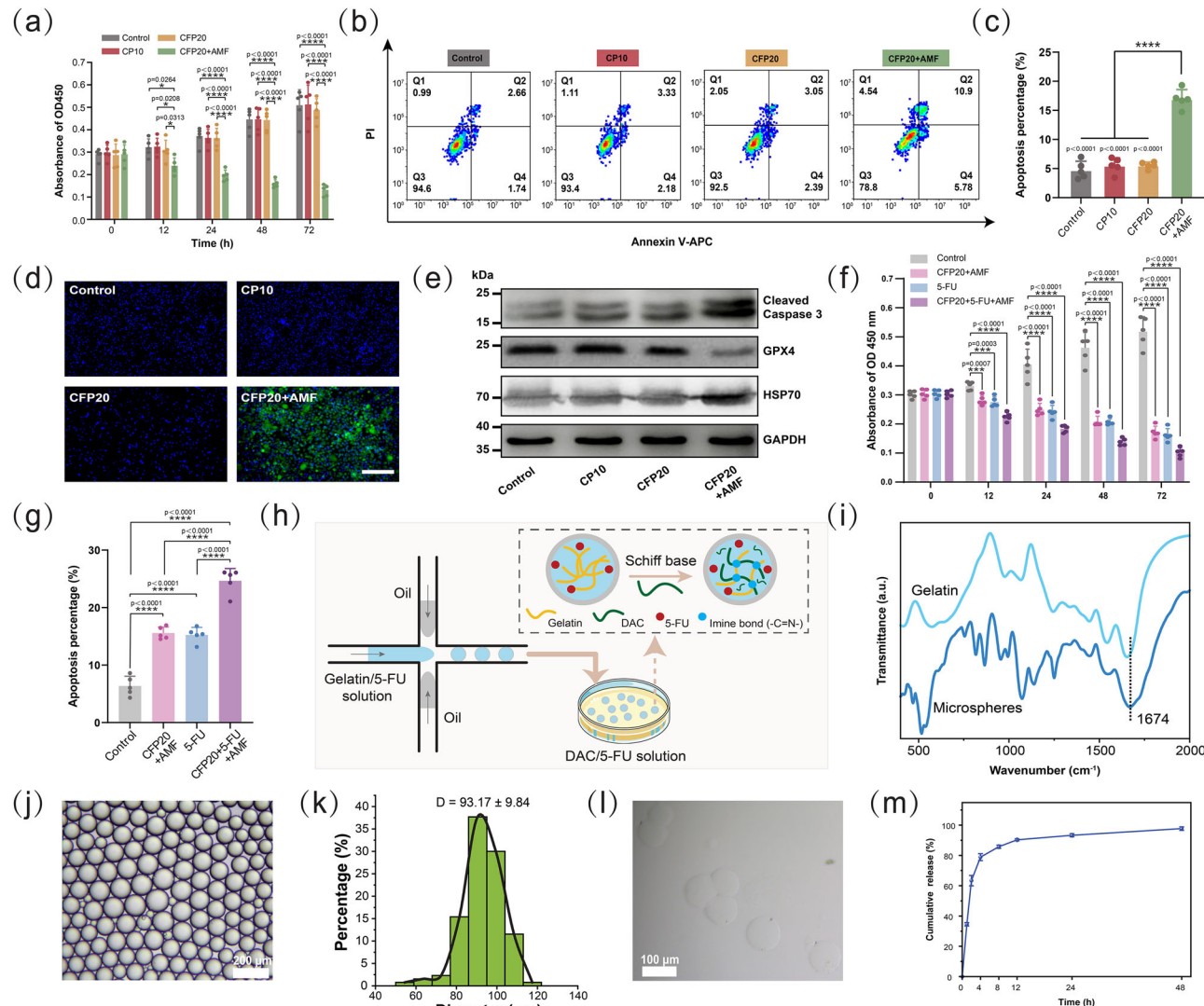

**Fig. 4 | In vitro anti-tumor effect of HTHSG. a** Cell viability of HCT-116 cells in the control, CP10, CFP20, and CFP20 + AMF groups as evaluated by the CCK-8 assay. The data were presented as the mean ± SD ($n = 5$ independent experiments). Statistical differences were analyzed by one-way ANOVA first, and then by Tukey's post hoc test. (*$p < 0.05$ and ****$p < 0.0001$). **b, c** Apoptosis of HCT-116 cells in control, CP10, CFP20 and CFP20 + AMF groups analyzed by Annexin V-APC and PI staining. The apoptosis percentage (%) is the proportion of Q2 + Q4 compared to total cells. The data were presented as the mean ± SD ($n = 5$ independent experiments). Statistical differences were analyzed by a two-tailed unpaired Student's t-test. (****$p < 0.0001$). **d** Representative images of DAPI (blue) and DCFH-DA (green) stained HCT-116 cells to detect intracellular •OH across control, CP10, CFP20, and CFP20 + AMF groups. Scale bar: 100 μm. Each experiment was repeated five times independently with similar results. **e** Western blotting for intracellular Cleaved Caspase 3, GPX4, and HSP70 expression in the control, CP10, CFP20, and CFP20 + AMF group. **f** Cell viability of HCT-116 cells in the control, CFP20 + AMF, 5-FU and CFP20 + 5-FU + AMF groups via CCK-8 assay. The data were presented as the mean ± SD ($n = 5$ independent experiments). Statistical differences were analyzed by one-way ANOVA first, and then by Tukey's post hoc test. (***$p < 0.001$ and ****$p < 0.0001$). **g** Apoptosis percentage of HCT-116 cells in the control, CFP20 + AMF, 5-FU and CFP20 + 5-FU + AMF groups. The data were presented as the mean ± SD ($n = 5$ independent experiments). Statistical differences were analyzed by two-tailed unpaired Student's t-test. (****$p < 0.0001$). **h** Schematic diagram of the preparation of drug-loaded hydrogel microspheres using a microfluidic approach. **i** FTIR spectra of microspheres, gelatin, and DAC. **j** The images of the droplets. Scale bar: 200 μm. Each experiment was repeated three times independently with similar results. **k** The size distribution of the droplets. **l** The images of the hydrogel microspheres. Scale bar: 100 μm. Each experiment was repeated three times independently with similar results. **m** The release profile of 5-FU from the drug-loaded CFP20 hollow tube. The data were presented as the mean ± SD ($n = 6$ independent samples). Source data are provided as a Source Data file.

PDX model using tumor tissues from CRC patients. The experimental workflow is shown in Fig. 6a, and patient characteristics are summarized in Supplementary Table 2. PDX tumors successfully retained the morphology and protein expression profiles of the parent tumor samples (Fig. 6b). Each PDX tumor was divided into seven parts and implanted into seven additional mice, which were grouped and treated identically to the CDX model. Temperature changes at the tumor site were recorded using an infrared thermal camera. Tumor temperatures in Groups 1, 2, 4, and 5 showed minimal changes, while those in Groups 6 (CFP20 + AMF) and 7

(CFP20 + 5-FU MSs+AMF) increased significantly, reaching approximately 45 °C (Fig. 6c).

After treatment, tumor volumes and weights were measured (Fig. 6d–f). Both systemic chemotherapy (Group 5) and localized 5-FU delivery via CP20 hydrospongel (Group 4) inhibited tumor growth ($p < 0.05$), as did magnetothermal therapy using CFP20 (Group 6). However, CFP20 + 5-FU MSs with AMF (Group 7) exhibited the most significant anti-tumor effects across all PDX models. This effect was particularly pronounced in Patient 3 (P3), where the integrated treatment achieved a tumor shrinkage rate of approximately 90%, far

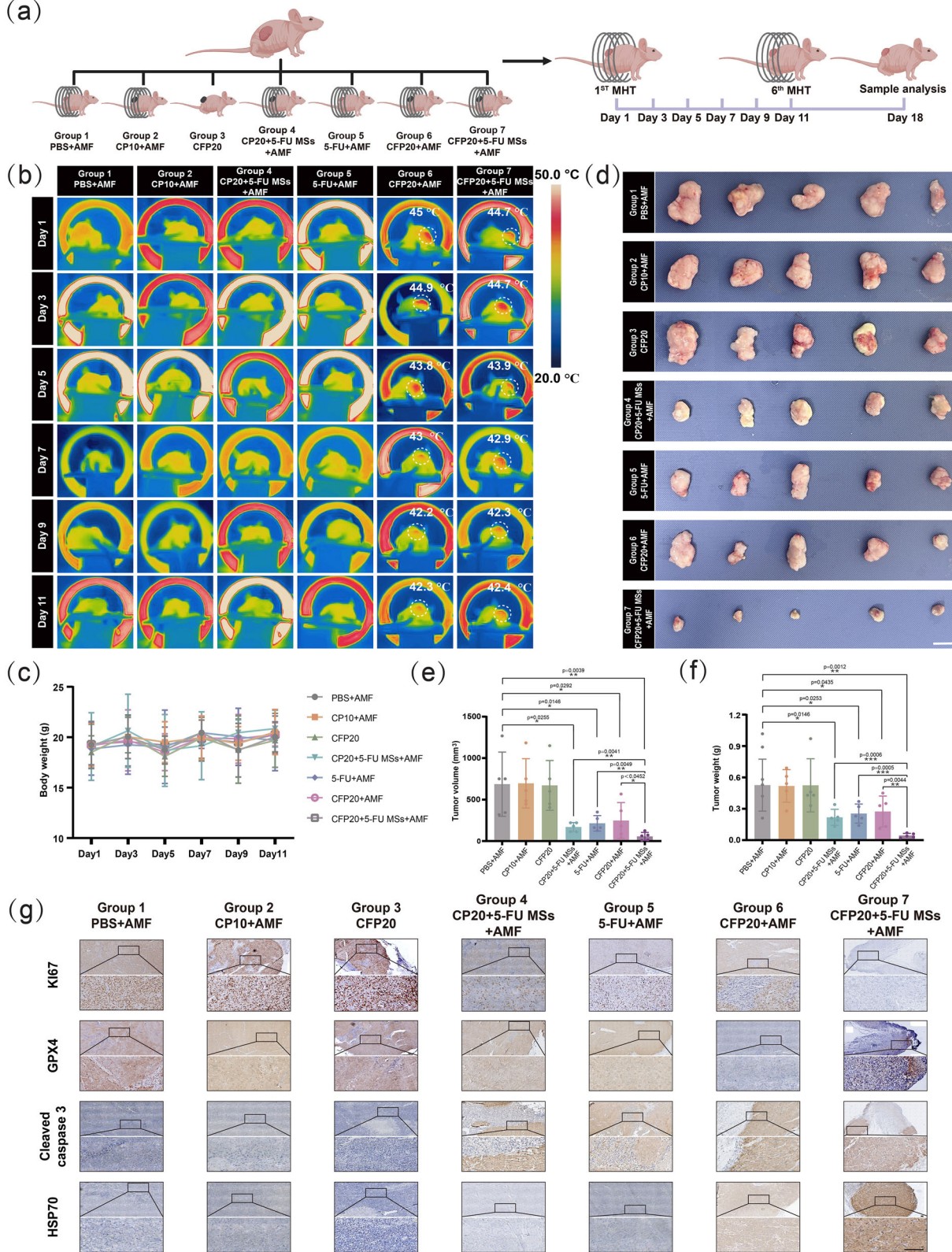

**Fig. 5 | In vivo establishment of the CDX model and the anti-tumor efficacy of HTHSG. a** Schematic of CDX model establishment and treatment. Created in BioRender. Zhao, J. (2025) https://BioRender.com/wo5g5bl. **b** Thermal images of CDX tumor-bearing mice during AMF exposure. **c** Body weight changes in each group of the CDX model. The data were presented as the mean ± SD (n = 5 mice per group). **d** Digital images of excised tumors from the CDX model. Scale bar: 1 cm. **e** Tumor volumes from colorectal CDX model mice in Groups 1, 2, 3, 4, 5, 6, and 7. The data were presented as the mean ± SD (n = 5 mice per group). Statistical

differences were analyzed by a two-tailed unpaired Student's t-test. (*p < 0.05, **p < 0.01). **f** Tumor weights from colorectal CDX model mice in Group 1, 2, 3, 4, 5, 6 and 7. The data were presented as the mean ± SD (n = 5 mice per group). Statistical differences were analyzed by two-tailed unpaired Student's t-test. (*p < 0.05, **p < 0.01 and ***p < 0.001). **g** Immunohistochemical analysis of KI67, Cleaved Caspase 3, GPX4 and HSP70 of tumor slices in different groups. Scale bar: 100 μm. Each experiment was repeated five times independently with similar results. Source data are provided as a Source Data file.

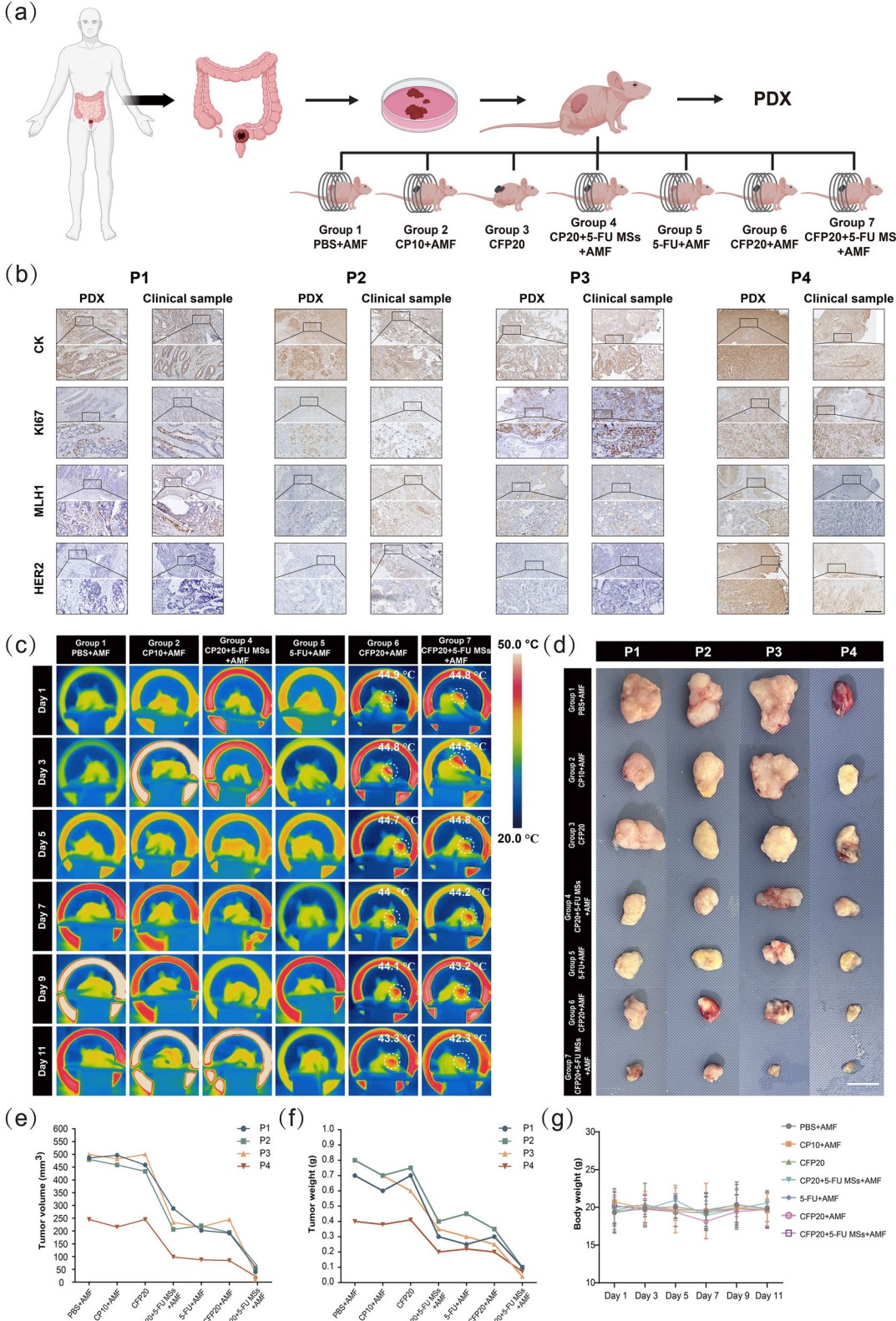

**Fig. 6 | In vivo anti-tumor efficacy and biocompatibility of HTHSG in a colorectal cancer PDX model. a** Schematic of PDX model establishment and treatment. Created in BioRender. Zhao, J. (2025) https://BioRender.com/37om8bc. **b** Immunohistochemical analysis for CK, KI67, MLH1, and HER2 in tumor slices from colorectal cancer patients and the PDX model to evaluate their similarity. Scale bar: 100 μm. **c** Thermal images of PDX tumor-bearing mice during AMF exposure.

**d** Digital images of the excised tumor of the PDX model. Scale bar: 1 cm. **e** Tumor volumes of colorectal PDX model mice for 4 patients in Groups 1, 2, 3, 4, 5, 6 and 7. **f** Tumor weights from different colorectal PDX model mice for 4 patients in Group 1, 2, 3, 4, 5, 6 and 7. **g** Body weight changes across different mouse groups of the PDX model. The data were presented as the mean ± SD (*n* = 4 mice per group). Source data are provided as a Source Data file.

surpassing the effects of single therapies. Detailed group comparisons for the PDX models are provided in Supplementary Table 3. Body weight remained stable across all groups throughout the treatment, further indicating the safety of HTHSG (Fig. 6g). Immunohistochemical results were consistent with findings from the CDX model, showing reduced KI67 expression and increased Cleaved Caspase 3 in treatment groups (Supplementary Fig. 20). H&E staining of major organs from all mice receiving PDX tumors from four patients (P1-P4) revealed no evidence of systemic toxicity, confirming the biocompatibility of CFP20 + 5-FU MSs (Supplementary Figs. 21–24).

Given the heterogeneity of malignant tumors, even those with similar origins may exhibit differing treatment sensitivities[57]. In this study, while PDX tumors from all four CRC patients responded to chemotherapy or magnetothermal therapy alone, the integrated treatment with CFP20 + 5-FU MSs consistently demonstrated superior and stable efficacy. Especially, the tumor from Patient 3, located near the anus, responded exceptionally well to the multimodal approach, highlighting its potential benefits in anal preservation and quality-of-life improvements for patients with distal CRC. These results suggest that HTHSG, integrating magnetothermal, chemo-, and chemodynamic therapies, provides an innovative and effective treatment regimen for preoperative CRC management.

## Anti-tumor effect of HTHSG in orthotopic mice models

Furthermore, an orthotopic mice model was established to evaluate the anti-tumor efficacy and biosafety of HTHSG under conditions that closely mimic the in vivo environment of CRC (Fig. 7a). MC38-luc cells were injected into the rectal walls of C57BL/6 mice to induce tumor formation. Once the bioluminescence intensity of the tumors reached $1 \times 10^6$ p/s/cm$^2$/sr, the mice were randomly divided into seven treatment groups, similar to the CDX model (Fig. 7b). These included PBS + AMF, CP10 + AMF, CFP20, CP20 + 5-FU MSs+AMF, 5-FU + AMF, CFP20 + AMF, and CFP20 + 5-FU MSs+AMF. Hydrospongels (25 mg) were implanted rectally around tumors in Groups 2, 3, 4, 6, and 7, and mice in AMF-treated groups were exposed to AMF for 10 minutes on days 1, 3, 5, 7, 9, and 11 (Supplementary Movie 2). Infrared thermal imaging confirmed that the local tumor temperature in Groups 6 and 7 increased to ~45 °C under AMF, while no significant temperature changes were observed in the other groups (Fig. 7c).

Seven days after the final AMF treatment, tumor bioluminescence was evaluated using a small animal imaging system after d-luciferin injection. Groups treated with CP20 + 5-FU MSs+AMF, 5-FU + AMF, or CFP20 + AMF (Groups 4, 5, and 6) showed significantly lower tumor bioluminescence compared to the PBS + AMF group. Especially, Group 7 (CFP20 + 5-FU MSs+AMF) demonstrated the lowest bioluminescence intensity, significantly outperforming Groups 4, 5, and 6 (Fig. 7d). CT imaging revealed that the hydrospongels did not exert pressure on or cause damage to surrounding normal tissues (Fig. 7e).

Following imaging, the mice were sacrificed, and colorectal tumors (together with colorectum) were collected for further analysis (Fig. 7f, Supplementary Fig. 25). Tumor volume measurements corroborated the bioluminescence results, with Group 7 showing the greatest tumor suppression ($p < 0.0001$). Comprehensive comparisons of tumor data across all groups are summarized in Supplementary Table 4. Body weight remained stable among all groups throughout the treatment period, suggesting favorable systemic tolerance (Supplementary Fig. 26). Immunohistochemistry (Fig. 7g) revealed no significant differences in KI67, GPX4, Cleaved Caspase 3, or HSP70 expression between Groups 1, 2, and 3. However, Groups 4, 5, and 6 exhibited reduced KI67 expression and increased Cleaved Caspase 3 levels. Groups 6 and 7 additionally showed reduced GPX4 and elevated HSP70 expression, consistent with the combined effects of thermal ablation and CDT. Group 7 demonstrated the most pronounced reductions in KI67 and elevations in Cleaved Caspase 3, confirming the superior efficacy of the integrated treatment. Flow cytometry analysis of spleen immune cell populations, including dendritic cells (DCs), CD4 + T cells, CD8 + T cells, and regulatory T cells (Tregs), showed no significant changes across groups, indicating no adverse effects on the immune system (Fig. 7h-k, Supplementary Figs. 27–30). Additionally, H&E staining of major organs revealed no signs of damage, further validating the biocompatibility of HTHSG (Supplementary Fig. 31).

This orthotopic model allowed us to closely simulate the disease environment of CRC, and provided valuable insights into the efficacy and safety of HTHSG in the colorectal tract. The results confirmed that HTHSG, integrating thermal ablation, chemotherapy, and CDT, significantly reduced tumor size compared to conventional treatments. Furthermore, CT imaging and histological analyses demonstrated the excellent biosafety of the hydrospongel. For distal CRC patients, localized application of HTHSG before surgery could effectively downstage tumors and improve the likelihood of anal preservation, offering a promising approach for preoperative CRC treatment.

## Feasible application of HTHSG

To evaluate the feasibility of applying HTHSG in larger mammals, we conducted experiments in a beagle using the CFP20 + 5-FU MSs hydrospongel. After bowel preparation, a guidewire was introduced to the designated position in the colorectum (Fig. 8a) at 10 cm proximal the anus and near the internal anal sphincter, a location that is critical in determining whether anal preservation is feasible during radical resection for cancer. Under anesthesia and with colonoscopic guidance, HTHSG was placed between the gastric and esophageal sacs of a Sengstaken-Blakemore Tube, which was then positioned along the guidewire (Fig. 8b–d). The gastric sac was inflated to fix the hydrospongel in the designated position (Supplementary Movie 3). We also designed a 3D-printed PLA bracket (Fig. 8e). The bracket has two parts, and HTHSG was placed in the groove of part 1 (Fig. 8e). The two parts of the bracket can be twisted together to be placed between the gastric and esophageal sac. The bracket could ensure the structural integrity of HTHSG and facilitate smooth placement during delivery. (Fig. 8f, Supplementary Movie 3). The Sengstaken-Blakemore Tube, first described in 1950[58], has been traditionally used for treating hemorrhaging esophageal varices via a balloon tamponade technique[59]. In our study, we leveraged the space between the gastric and esophageal sacs of the Sengstaken-Blakemore Tube to facilitate the fixed placement of HTHSG through colonoscopy. Additionally, when used in the colorectal tract, the Sengstaken-Blakemore Tube also demonstrated a secondary benefit in alleviating obstruction, functioning similarly to a transanal decompression tube[60].

The dog was exposed to an AMF for 10 minutes on days 1, 3, and 5 (Fig. 8g, Supplementary Movie 3). Before exposure, we confirmed the temperature change of HTHSG under the AMF by an infrared thermal camera (Fig. 8h, Supplementary Movie 3). Following the third treatment, a CT scan revealed no signs of pressure or damage to surrounding tissues, as evaluated by a radiologist (Fig. 8i). The hydrospongel was subsequently removed through colonoscopy under anesthesia. The dog remained in good health, exhibiting normal defecation post-treatment (Supplementary Movie 3).

Two days after HTHSG removal, defecography was performed to assess anal function. The result showed a dilated rectum and closed anal canal before defecation, as well as a contracted rectum and open anal canal after defecation. The results were similar to those found by defecography performed before any treatment (Fig. 8j, k). Pathological analysis, including Masson trichrome staining, showed no atrophy or collagen deposition[61] in the internal or external anal sphincters, further supporting preserved anal function (Fig. 8l, m). H&E staining of major organs revealed no signs of damage, confirming the hydrospongel's biocompatibility in the beagle dog model (Fig. 8n).

These findings demonstrate the safety and feasibility of HTHSG application in larger mammals, laying the groundwork for human

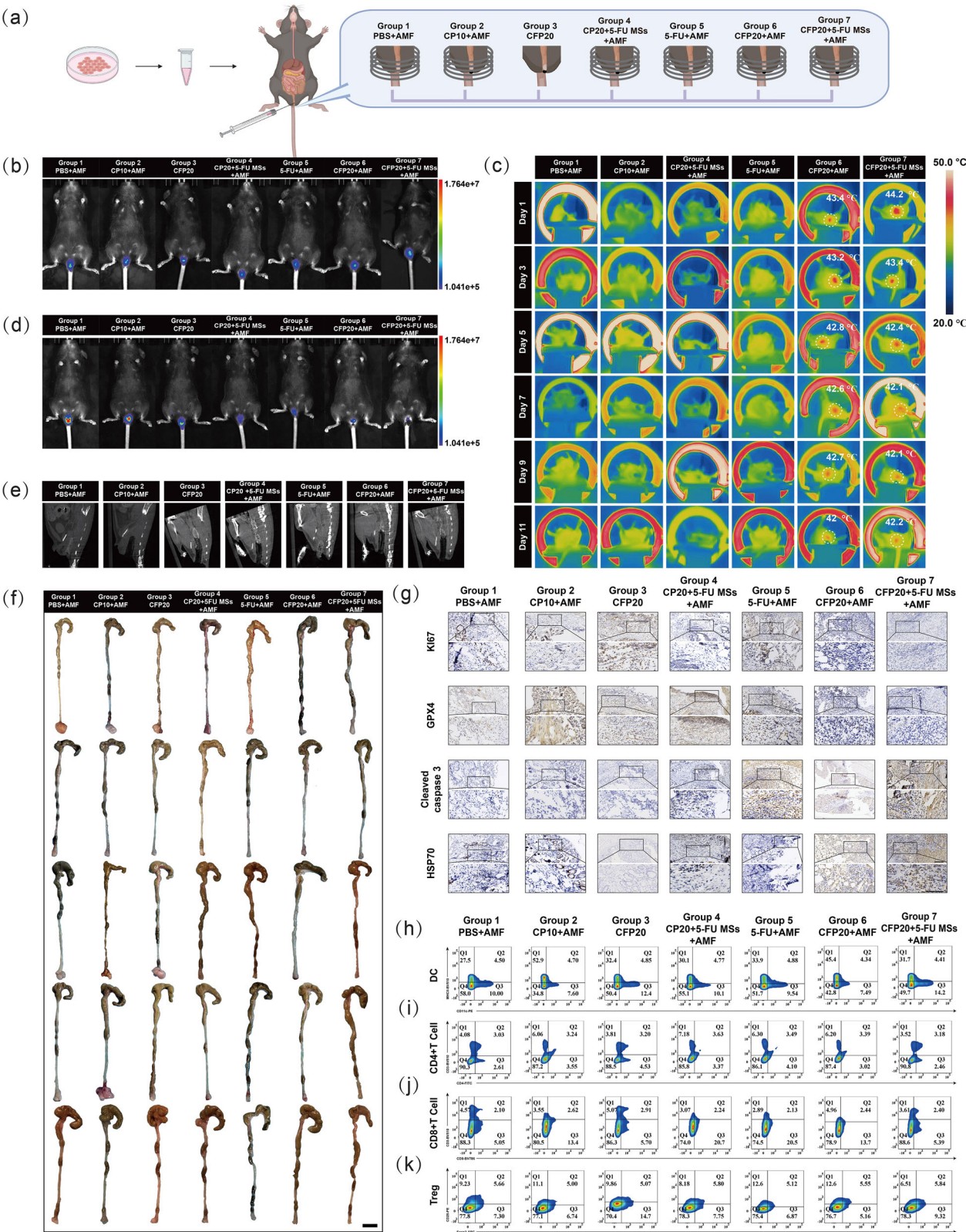

**Fig. 7 | Anti-tumor effects of HTHSG in orthotopic mouse models. a** Schematic of orthotopic rectal cancer mouse model establishment and treatment. Created in BioRender. Zhao, J. (2025) https://BioRender.com/9unaydd. **b** In vivo bioluminescence images of orthotopic rectal cancer mouse models before treatment. **c** Thermal images of orthotopic rectal cancer mice during AMF exposure. **d** In vivo bioluminescence images of orthotopic rectal cancer mice models after treatment. **e** In vivo CT images of orthotopic rectal cancer mice models during treatment. Each experiment was repeated five times independently with similar results. **f** Digital images of excised tumors from orthotopic rectal cancer mouse models. Scale bar: 1 cm. **g** Immunohistochemical analysis of KI67, Cleaved Caspase 3, GPX4 and HSP70 on tumor slices from different groups. Scale bar: 100 μm. Each experiment was repeated five times independently with similar results. Representative flow cytometry plots of DCs (**h**), CD4 + T cells (**i**), CD8 + T cells (**j**) and Treg cells (**k**) in the spleen after different treatments.

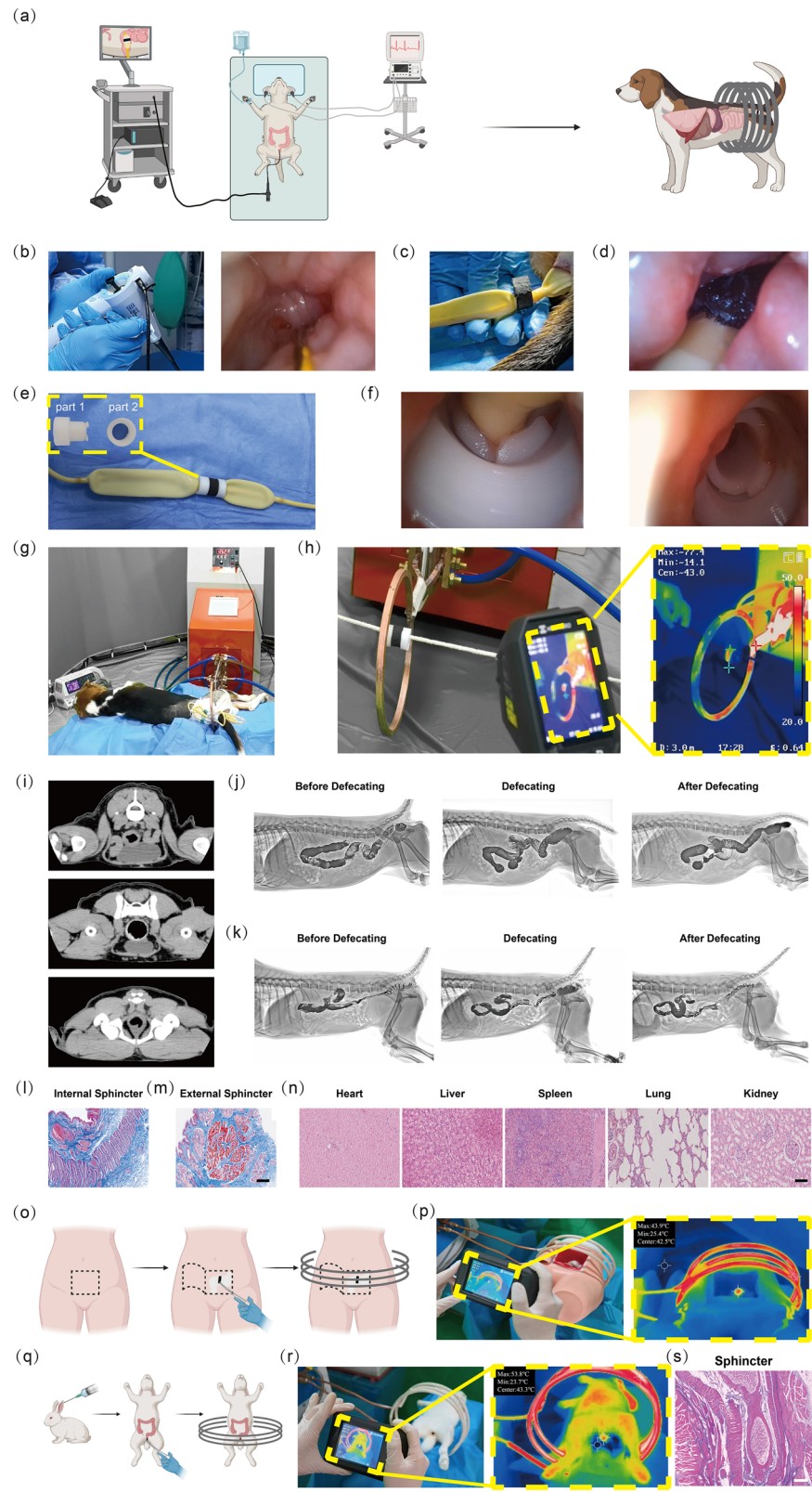

trials. The preserved anal function and lack of organ damage suggest that preoperative HTHSG application could offer hope for anal preservation in distal CRC patients. By reducing tumor size preoperatively, HTHSG may enhance surgical outcomes and patient quality of life.

To further evaluate the feasibility of HTHSG for human application, we conducted experiments using a dummy enema training model with dimensions comparable to those of a human. An AMF coil with a 32 cm diameter was constructed to accommodate the dummy, and HTHSG was placed in the colorectum before being exposed to the AMF (Fig. 8o). Infrared thermal imaging confirmed a temperature increase in HTHSG to approximately 42 °C under these conditions (Fig. 8p). To further assess safety, we implanted HTHSG rectally in a rabbit and subjected it to the same AMF and coil parameters as the dummy

**Fig. 8 | Feasible application of HTHSG. a** Schematic of HTHSG hydrospongel implantation and treatment in a beagle. Created in BioRender. Zhao, J. (2025) https://BioRender.com/juwc1cn. **b** Digital image of the guidewire placed in the designated position. **c** Digital image of the HTHSG hydrospongel placed into a Sengstaken-Blakemore Tube. **d** Digital image of HTHSG and Sengstaken-Blakemore Tube placed at the designated position in the dog's bowel. **e** Digital image of HTHSG (with 3D-printed PLA bracket) and Sengstaken-Blakemore Tube. **f** Digital image of HTHSG with a bracket placed at the designated position in the dog's bowel. **g** Digital image of a beagle exposed to AMF. **h** Thermal image of HTHSG during AMF exposure. **i** In vivo CT images of a beagle during treatment. **j** Defecation imaging of a beagle before treatment. **k** Defecation imaging of a beagle after

treatment. Representative Masson trichrome staining images of the beagle internal anal sphincter (**l**) and external anal sphincter (**m**). Scale bar: 0.5 mm.
**n** Representative H&E staining images of beagle major organs, including heart, liver, spleen, lung, and kidney of beagle. Scale bar: 100 μm. **o** Schematic of HTHSG hydrospongel implantation and treatment in a dummy enema training model. Created in BioRender. Zhao, J. (2025) https://BioRender.com/yanxf85. **p** Thermal images of dummy enema training model during AMF exposure. **q** Schematic of HTHSG hydrospongel implantation and treatment in a rabbit. Created in BioRender. Zhao, J. (2025) https://BioRender.com/gomd4sd. **r** Thermal images of rabbit during AMF exposure. **s** Representative Masson trichrome staining images of rabbit sphincter. Scale bar: 0.5 mm.

(Fig. 8q). The rabbit underwent 30-minute exposures on days 1, 3, and 5, with temperature changes monitored by infrared thermal imaging (Fig. 8r). Histological analysis (Masson trichrome staining) revealed no atrophy or collagen deposition in the anal sphincters (Fig. 8s), while H&E staining of major organs showed no signs of tissue damage (Supplementary Fig. 32), confirming the biocompatibility of HTHSG under these conditions.

In our study, HTHSG was exposed to AMF across three models of varying size: mice, beagles, and the human-sized dummy. Each model required appropriately sized AMF coils. Specifically, as the coil diameter increased, achieving effective magnetothermal performance became more challenging under identical AMF conditions[62,63]. Therefore, different equipment was used for each model to ensure HTHSG could reliably exhibit a magnetothermal effect. In addition, as demonstrated in the dummy experiments, larger models—such as humans—may require longer exposure times to reach the desired therapeutic temperature. The specific AMF parameters (strength and frequency) utilized for mice, beagle, and dummy experiments are summarized in Supplementary Table 5. Importantly, the product of frequency and field strength in all cases remained below the accepted biological safety limit of $5 \times 10^9$ A/m/s[64]. All AMF settings used in our study met this safety criterion.

In summary, our data demonstrate that the magnetothermal effect of HTHSG can be achieved across a range of model sizes by adapting AMF coil size and exposure time. The safety and biocompatibility of these conditions were further supported by histological analysis. For human application condition, the safety of treatment was verified in a rabbit. However, we acknowledge that before clinical application in humans, the long-term safety of this approach will require further investigation.

We also conducted a comprehensive review of hydrogel and nanoparticle-based delivery systems for CRC treatment (Supplementary Table 6)[14,15,65–73]. Compared with previous studies, HTHSG presents distinct advantages by integrating magnetothermal, chemo-, and chemodynamic therapies into a single therapeutic platform. Furthermore, unlike previous studies focusing solely on in vitro or small-animal models, this study validates HTHSG's efficacy in both PDX models and a large-animal feasibility study, thus bridging the gap toward clinical application. Although these findings are encouraging, several limitations of this study warrant further investigation. First, a systematic assessment of HTHSG's long-term biocompatibility is essential to confirm its safety profile. Furthermore, optimizing AMF parameters may further improve the tumor downstaging efficacy in patients. Moreover, the clinical adoption of HTHSG necessitates investment in AMF-generating devices and interdisciplinary collaboration to refine colonoscopic implantation techniques.

In conclusion, this study introduces a hollow-tube-like hydrospongel (HTHSG) as an innovative preoperative treatment for advanced-stage colorectal cancer (CRC). Engineered from cellulose nanofibers and Fe₃O₄@PDA nanoparticles, HTHSG exhibits rapid swelling (~10 s), high mechanical strength (>250 kPa), and a biomimetic tubular structure tailored for colorectal delivery. Its triple synergistic mechanism—magnetic hyperthermia-mediated ablation,

Fenton reaction-driven chemodynamic therapy (CDT), and controlled 5-FU release—enhances antitumor efficacy, generating a cascade amplification effect. In vitro, HTHSG reduced cancer cell viability by 37.5% more than conventional chemotherapy, while increasing apoptosis by 61.6%. In vivo, it demonstrated >85% tumor inhibition in multiple preclinical models, with a >30% improvement over standard chemotherapy in PDX models. Successful Beagle implantation and human-sized dummy experiment further confirmed its clinical feasibility. These findings highlight HTHSG as a promising neoadjuvant therapy for CRC, particularly for distal tumors requiring anal preservation, warranting further clinical evaluation for broader application.

## Methods
### Ethical statement
Colorectal cancer tissue samples used for generating PDX models were obtained from CRC-diagnosed patients in the First Hospital of China Medical University, and the procedure was in strict compliance with the Research Ethics Committee of the First Hospital of China Medical University (Approval No.:2023-245) and the principles of the Declaration of Helsinki. Informed consent was obtained from all patients participating in the study prior to sample collection and obtaining clinical information. Female BALB/c nude mice (6–8 weeks and ~20 g) and female C57BL/6 mice (6–8 weeks and ~20 g) were purchased from GemPharmatech Co., Ltd (China). All the mice were housed in a room with a temperature controlled at 25 °C, following a 12 h dark/12 h light cycle, and with free access to food and water. The female beagle dog (~27 months and 15 Kg) was purchased from Shenzhen TOP Biotechnology Co., Ltd. The female New Zealand white rabbits (~6 months and 2.5 Kg) were purchased from Kondebio Co., Ltd (China). All experimental procedures involving animals in this study were approved by the Research Ethics Committee of China Medical University (Permit No.: KT2022470; KT20241798; CMU20251379) according to the National Research Council's Guide for the Care and Use of Laboratory Animals. The maximum tumor burden permitted was 2000 mm³, and the maximal tumor size/burden in this study was not exceeded.

### Materials
Ferrous chloride tetrahydrate (FeCl₂·4H₂O), ferric chloride hexahydrate (FeCl₃·6H₂O), and ammonium hydroxide (NH₄OH) 25% solution were purchased from Sinopharm Chemical Reagent Co., Ltd (China). Tris (hydroxymethyl) aminomethane (Tris), dopamine hydrochloride (DA), 3,3′,5,5′-tetramethylbenzidine dihydrochloride hydrate (TMB), hydrogen peroxide solution, lysozyme (≥20000 U/mg), and 5-fluorouracil (5-FU) were obtained from Aladdin (China). Gelatin was purchased from Aladdin (China), fluorinated oil, and emulsion breaker were purchased from Fluidiclab (China). Carboxylated cellulose nanofiber (CNF-C, diameter: 4–10 nm, length: 200 nm) was purchased from Macklin (China), and 1-ethyl-3-(3-dimethyl aminopropyl) carbodiimide hydrochloride (EDC) and N-Hydroxysuccinimide (NHS) were purchased from Sigma-Aldrich (U.S.A.). McCoy's 5 A medium and fetal bovine serum (FBS) were purchased from Procell Life Science & Technology (China). Phosphate-

buffered solution (PBS) was purchased from GeneZe (China). The Cell Counting Kit-8 (CCK-8) was purchased from Dojindo Laboratories (Japan). The Total Protein Extraction Kit, bicinchoninic acid (BCA) kit, SDS-polyacrylamide gel electrophoresis (SDS-PAGE), and Annexin V-APC/PI apoptosis detection kit were purchased from KeyGen Biotech (China). Polyvinyl difluoride (PVDF) membranes were purchased from Millipore (U.S.A.). Sodium citrate buffer was purchased from MXB Biotechnologies (China). 2'7'-dichlorofluorescein diacetate (DCFH-DA) was purchased from Sigma-Aldrich (U.S.A.). Dimethyl sulfoxide (DMSO) and EDTA-free trypsin were procured from Solarbio (China). Glyceraldehyde 3-phosphate dehydrogenase monoclonal antibody (GAPDH, 60004-1-Ig) and HSP70 polyclonal antibody (10995-1-AP) were purchased from Proteintech (China). An anti-KI67 antibody (KI67, ab15580), anti-glutathione peroxidase 4 antibody (GPX4, ab125066), anti-pan Cytokeratin antibody (CK, ab7753), and hematoxylin were purchased from Abcam (U.K.). Human epidermal growth factor receptor 2 monoclonal antibody (HER2, 40419) was purchased from Signalway Antibody (U.S.A.). MutL homolog 1 monoclonal antibody (MLH1, YM0443) was purchased from Immunoway Biotechnology Company (China), and anti-Cleaved Caspase 3 antibody (Cleaved Caspase 3, 9446 s) was purchased from Cell Signaling Technology (U.S.A.). Goat anti-mouse IgG antibody, goat anti-rabbit IgG antibody, 3,3 N-Diaminobenzidine Tetrahydrochloride (DAB), and ABC (Avidin-Biotin Complex) Kits were provided by Vector Laboratories (U.S.A.). Eosin solution and hydrochloric acid ethanol (HCl/EtOH) were purchased from Servicebio (China).

## Synthesis of $Fe_3O_4$ and $Fe_3O_4$@PDA NPs

$Fe_3O_4$ NPs were synthesized using the co-precipitation method. In brief, 0.6958 g of $FeCl_2 \cdot 4H_2O$ and 1.8921 g of $FeCl_3 \cdot 6H_2O$ were added to 150 mL of deionized water and stirred at ambient temperature under a nitrogen atmosphere until completely dissolved. 10 mL of ammonium hydroxide was then rapidly added to the solution and stirred at 80 °C for 1 h. After the solution was cooled to room temperature, precipitated particles were collected using a permanent magnet and washed three times with deionized water, followed by lyophilization to obtain $Fe_3O_4$ NPs.

The as-prepared $Fe_3O_4$ NPs (10 mg) were added to 50 mL of Tris-HCl solution (10 mM, pH=8.5) and sonicated for 10 min to disperse homogeneously. Subsequently, 10 mg of DA was added to the solution and stirred in the dark. After 8 h, the NPs were collected using a magnet, washed three times with deionized water, and then lyophilized to obtain $Fe_3O_4$@PDA NPs. Pure PDA NPs without $Fe_3O_4$ were prepared using the same method.

## Characterization of $Fe_3O_4$ and $Fe_3O_4$@PDA NPs

The UV-Vis spectra of nanoparticle aqueous solutions (1 mg/mL) were measured by a microplate reader (CLARIOstar, BMG, Germany) to determine the absorption capacity in the near-infrared (NIR) region. The morphology of the NPs was observed using a transmission electron microscope (TEM, JEM-2100F, JEOL, Japan), and the hydrodynamic size and zeta potential were measured using a zetasizer (Nano ZS90, Malvern, UK). Fourier transform infrared spectra (FTIR) of the NPs were collected using an FTIR spectrometer (Nicolet iS50, Thermo Fisher Scientific, U.S.A.). The X-ray diffraction (XRD) patterns of NPs were recorded using an XRD spectrometer (SmartLab SE, Rigaku, Japan) using Cu Kα radiation. The X-ray photoelectron spectra (XPS) were acquired using an XPS spectrometer (ESCALAB 250Xi, Thermo Fisher Scientific, U.S.A.) with monochromatic Al Kα radiation. The magnetic properties of the NPs were characterized by a superconducting quantum interference device-vibrating sample magnetometer (SQUID-VSM, MPMS3, Quantum Design, U.S.A.) at a temperature of 300 K.

## Reactive oxygen species (•OH) generation assessment

TMB oxidation assay was performed to evaluate the generation of •OH by the Fenton reaction between NPs and $H_2O_2$. In brief, 1 mg of NPs was added to 4 mL of PBS solution (pH = 6.0) containing TMB (0.5 mg/mL) and $H_2O_2$ (1 mM). After incubation for 60 min at 37 °C, the solution was centrifuged, and the absorbance of the supernatant at 665 nm was measured using a microplate reader. PBS solution without NPs or $H_2O_2$ served as the control.

## Preparation of hydrospongels and HTHSG

Hydrospongel was prepared using $Fe_3O_4$@PDA NPs crosslinked with CNF-C. Prepared $Fe_3O_4$@PDA NPs were added to 5 mL of deionized water and sonicated for 5 minutes to make them well dispersed. CNF-C (0.2 g), EDC (0.115 g), and NHS (0.023 g) were then added to the solution, stirred evenly, and injected into the mold (cube with 10 mm side length). The carboxylate groups (-COOH) in CNF-C and the amine groups (-$NH_2$) in $Fe_3O_4$@PDA NPs were utilized to create amide bonds in the presence of EDC and NHS, producing a chemically crosslinked hydrogel. All steps were carried out in an ice bath to avoid rapid crosslinking reactions. After 12 hours, the cross-linked hydrogels were lyophilized at −60 °C for 24 h using a vacuum freeze dryer (FD-150, Biocool, China). The lyophilized hydrospongels were then swelled in deionized water for 5 minutes and lyophilized again to remove residual water-soluble reagents. The lyophilized HTHSG was prepared following the same procedures except for using a 3D-printed mold (Supplementary Fig. 6).

## Characterization of hydrospongels

The swelling capacity was assessed by recording the weight change of the lyophilized hydrospongels in PBS (pH = 7.4) at 37 °C for 1 h. The mechanical properties of the swollen hydrospongels were analyzed using a universal testing system (Instron 5944, U.S.A.), and the compressive modulus and fracture toughness were determined using the stress-strain curve. To investigate the hydrospongels' magnetothermal properties with different nanoparticle concentrations, swollen hydrospongels were placed in an alternating magnetic field (AMF, 1.3 kA/m, 255 kHz) for 10 min. The infrared thermal images and temperature change ($\Delta T$) were recorded using an infrared thermography camera (FLIR ONE Edge, U.S.A.).

## Characterization of HTHSG

The FTIR spectra of HTHSG were collected using an FTIR spectrometer, and the morphology of HTHSG was performed on scanning electron microscopy (SEM, Regulus 8100, Hitachi, Japan). The elemental composition and distribution of HTHSG were recorded by energy-dispersive X-ray spectroscopy (EDS, Octane Elect, EDAX, U.S.A.). The HTHSG's capacity to produce •OH by Fenton reaction was assessed using the TMB oxidation assay. Briefly, 10 mg of lyophilized HTHSG was placed into 4 mL of PBS solution (pH = 6.0) containing TMB (0.5 mg/mL) and $H_2O_2$ (1 mM) and incubated at 37 °C and 45 °C, respectively. The absorbance at 665 nm was measured at different endpoints (10, 30, and 60 min). In addition, the release of iron ions from HTHSG was further examined to determine the mechanism of the Fenton reaction. Briefly, 10 mg of lyophilized HTHSG was placed into 5 mL of PBS solution (pH = 6.0) and incubated at 37 °C for 7 days. Each day, HTHSG was removed and placed in AMF for 10 minutes, and then it was put back. Meanwhile, HTHSG without AMF treatment was served as the control. The iron ions content in the solution was detected using an inductively coupled plasma mass spectrometry (ICP-MS, Agilent 7800, U.S.A.), with three replicates in each group.

## COMSOL simulation of thermal conductivity of colorectal cancer

COMSOL Multiphysics 6.2 software was applied to simulate temperature conduction. The initial tissue temperature was set at 37 °C, and a constant thermal condition of 45 °C was applied at the boundary of the unembedded region of the tumor model. In constructing the bowel cancer model, the tumor dimensions were 5 cm × 5 cm × (1 cm in the

central region or 1.3 cm in the marginal region), and the intestinal wall dimensions were 6 cm × 6 cm × 0.5 cm. The biological tissue properties were provided by the built-in COMSOL database. The thermal conductivity of the tumor tissue was set at 0.5 W/m·K and the thermal conductivity of the intestinal wall was 0.542 W/m·K.

## Preparation and characterization of drug-loaded hydrogel microspheres

The drug-loaded hydrogel microspheres were prepared using a template of water-in-oil (W/O) single emulsion and generated in the polydimethylsiloxane microfluidic device with flow-focusing junction (PDMS-FF-100, Fluidiclab, China). Briefly, 5-FU (10 mg/mL) was first dissolved into a 10-fold PBS solution (pH = 7.4) under ultrasound, and gelatin (4% w/v) was then dissolved therein at 37 °C to serve as the dispersed phase. In addition, the fluorinated oil (Drop-Surf droplet generation oil, FluidicLab, China) was used as the continuous phase and all phases were injected into the microchannels at constant flow rates (0.1 mL/min) controlled by syringe pumps (LSP01-1Y, Rongbai Pump, China). The fabrication process was observed using a microscope (MMJ31, Microtomo, China), and the generated 5-FU-loaded gelatin emulsion droplets were collected with dialdehyde cellulose (DAC) aqueous solution (2% w/v) containing 5-FU (10 mg/mL) in a petri dish and immersed for 1.5 hours to allow cross-linking. The resulting hydrogel microspheres were subjected to an emulsion breaker (Drop-Surf demulsifier, FluidicLab, China) for 5 minutes, followed by washing with deionized water and collection by centrifugation a centrifuge (TD5, Yingtai Instrument, China) at 67 × g for 5 minutes for subsequent assays.

The hydrogel microspheres were lyophilized using a freeze dryer, and the FTIR spectra of microspheres, gelatin, and DAC were recorded using an FTIR spectrometer to assess the cross-linking of microspheres.

## Drug content calculation in 5-FU loaded microspheres

In brief, 0.25 g of microspheres were placed in a centrifuge tube containing 10 mL of PBS and incubated at 37 °C for 48 hours to degrade completely. Subsequently, the absorbance of the solution at 266 nm was measured using a UV-vis spectrophotometer (AQ8100, Thermo Orion, U.S.A.). The drug content of the microspheres was calculated from the absorbance and the standard curve, and hence the mass fraction of the drug in the microspheres.

## Degradation

The lyophilized HTHSG was weighed ($W_d$), then immersed in PBS (pH=7.4) containing lysozyme (1 mg/mL) and incubated at 37 °C. Every 2 days, the solution was replaced with a fresh one, and the HTHSG was placed in an AMF for 10 min. The remaining mass of the HTHSG was weighed ($W_s$) after freeze-drying, and the weight change was calculated by the following equation (Eq. 1):

$$\text{Weight change}(\%) = (W_s - W_d)/W_S \times 100\% \qquad (1)$$

## Drug loading and release

The 5-FU solution (5 mg/mL in deionized water) was dropwised into a lyophilized HTHSG to obtain a drug-loaded HTHSG. The drug-loaded HTHSG was then immersed in 40 mL of PBS solution at 37 °C, and the PBS solution was replaced with fresh one at different endpoints. The absorbance of the releasing medium at 266 nm was measured using a UV-Vis spectrophotometer (AQ8100, Thermo Orion, U.S.A.), and the release ratio of 5-FU was determined using a standard curve obtained by fitting the absorbances of seven known 5-FU concentrations.

The 5-FU microspheres-loaded HTHSG was prepared with the same method as the previously prepared HTHSG, except that 1.16 g of 5-FU-loaded hydrogel microspheres were added to 5 mL of deionized water simultaneously with Fe₃O₄@PDA NPs to realize drug loading.

The mass ratio of 5-FU in the hydrogel microspheres was 0.86%, while the 5-FU content of the HTHSG was 2 mg/mL. Briefly, the drug-loaded HTHSG was immersed in 30 mL of PBS solution (containing 20% serum) at 37 °C, and the release ratio of 5-FU was monitored.

Typically, 5 mL of the hydrogel precursor solution yields approximately 250 mg of hydrospongel. The drugs and compositions used in all animal studies and experimental groups are summarized in Supplementary Table 7.

## Cell culture

The HCT-116 cancer cell line (KGG3229-1), human intestinal epithelial cells NCM460 (KGG3113-1), and MC38 mouse colon cancer cell line (KGG2228-1) were purchased from KeyGen Biotech (China). All cell lines were authenticated using STR profiling, and were confirmed negative for mycoplasma. HCT-116 cancer cells were cultured in McCoy's 5 A medium supplemented with 10% FBS in an incubator containing 5% $CO_2$ at 37 °C. The NCM460 and MC38 cells were cultured in RPMI 1640 medium supplemented with 10% FBS in an incubator containing 5% $CO_2$ at 37 °C.

## Cell counting kit-8 (CCK-8) assay

The CCK-8 assay was used to determine cell viability. Briefly, the HCT-116 cells were seeded in a 96-well plate with a density of $1 \times 10^4$ cells per well. A 0.3 mg HTHSG sample was placed in each well. At different endpoints (0, 12, 24, 48, and 72 h), the CCK-8 reagent was added to the medium at a ratio of 1:10, then the optical density (OD) was determined 1 h later using a SpectraMax Absorbance Reader (Molecular Devices, U.S.A.) at a wavelength of 450 nm. Five parallel wells were analyzed in each group, and the experiment was repeated 3 times.

## Flow cytometry

HCT-116 cells were cultured in a 6-well plate with a density of $1 \times 10^6$ per well. A 6 mg HTHSG sample was placed in each well for 24 h. According to the manufacturer's instructions, apoptotic cells were stained using Annexin V-APC and propidium iodide. The apoptosis rate in HCT-116 cells was analyzed using a flow cytometer (Accuri C6, BD Biosciences, U.S.A.). The results were expressed as the percentage of apoptotic cells in total cells. The experiment was repeated 3 times.

## DCFH-DA fluorescent staining

The cells were seeded in a 6-well plate at a density of $1 \times 10^6$ cells/well, and a 6 mg HTHSG sample was added to the medium. After 12 h, 0.4 μL of DCFH-DA/DMSO solution (25 mg/mL) was added to the medium for cell staining. After 3 min, the cells were washed with PBS and stained with DAPI for 15 min. The fluorescence images were observed using an inverted microscope (DMi8, Leica, Germany) at excitation and emission wavelengths of 469 nm and 525 nm, respectively.

## Western blotting

Western blotting was used to detect the expression of intracellular proteins. In short, the total protein of HCT-116 cancer cells was extracted using a Total Protein Extraction kit. The protein concentration was determined using a BCA Protein Assay kit. The proteins were separated by SDS-polyacrylamide gel electrophoresis (SDS-PAGE) and transferred to a PVDF membrane. After being blocked using non-fat milk (5%), the membranes were incubated with specific primary antibodies overnight at 4 °C followed by secondary antibodies at room temperature for 1 h. Finally, the membranes were imaged using Amersham™ ImageQuant™ 800 Western blot imaging systems (Cytiva, U.S.A.). The GAPDH antibody was the control for total protein input. The western blotting analysis was repeated three times. The primary antibodies used were as follows: anti-glutathione peroxidase 4 (1:1000, Abcam), anti-Cleaved Caspase 3 (1:1000, Cell Signaling Technology), anti-HSP70 (1:1000, Proteintech) and anti-GAPDH (1:1000, Proteintech).

## GSH quantitation

GSH analysis was performed by UPLC-PDA equipment on Wthe aters ACQUITY system H-Class. GSH samples were derivatized by DTNB (5,5′-Disulfanediylbis). Coupled to a photodiode array detector, this system could monitor absorbance from derivatized GSH at 412 nm. Data acquisition and processing were performed using Empower 3 software (Waters Corporation, USA). The GSH from derivatized samples was analyzed as followed: chromatographic separation was performed using an ACQUITY UPLC BEH C18 column (130 Å, 1.7 µm, 2.1 × 50 mm, from WATERS) at 40 °C under an isocratic flow of 0.3 mL min$^{-1}$ containing 50% acetonitrile, 0.1% formic acid, 49.9% Milli-Q water.

## Cell-derived xenograft (CDX) and patient-derived xenograft (PDX) model preparation

For the CDX model preparation, $2 \times 10^6$ HCT-116 cells in 0.2 mL PBS were subcutaneously injected into the right buttock of the BALB/c nude mice. When the tumor tissue grew to 1000 mm$^3$, it was evenly subdivided into seven parts and implanted in another seven mice. Mice in Group 1 were intraperitoneally injected with 100 µL PBS (twice times a week). Mice in the Group 5 were intraperitoneally injected with 100 µL 5-FU solution (25 mg/kg, twice a week). For Group 2, 3, 4, 6, and 7, mice were administered with the hydrospongel once the tumor tissue grew to 1000 mm$^3$. Under anesthesia, the skin tissue around the tumor in the mice was cut open. Afterwards, the skin was separated from the tumor tissue, the hydrospongel was implanted, and the skin wound was closed using 3-0 non-absorbable sutures. Mice in Groups 1, 2, 4, 5, 6, and 7 were exposed to AMF (H = 1.3 kA/m, f = 255 kHz) after anesthesia on days 1, 3, 5, 7, 9, and 11, and the temperature change at the tumor site was recorded using an infrared thermography camera. All mice were sacrificed on the 7$^{th}$ day after treatment, and tumor volume (V = $(length \times width^2)/2$) and tumor weight were recorded. The tumor tissues and major organs (heart, liver, spleen, lung, and kidney) were removed and fixed in a 4% formaldehyde solution for subsequent analyses.

For the PDX model, the tumor tissues from CRC patients were preserved in an ice-cold culture medium, and 0.3 gof tumor tissue was implanted subcutaneously into the right buttock of the BALB/c nude mice. The subsequent treatments were the same as in the CDX model preparation.

## Orthotopic colorectal cancer mice models construction

For constructing the orthotopic colorectal cancer mouse models, mice were anesthetized and kept in a supine position. The lower extremities of the mouse were then raised to the cranial side at a 10° angle to expose the rectum. 10 µL of the Luc-MC38 cell suspension ($5 \times 10^5$ cells per mouse) was carefully injected into the submucosal layer of the rectum. A swelling of the rectum wall can be seen after a successful injection. After injection, the mouse is placed in a clean cage to recover from anesthesia. Established orthotopic colorectal cancer mouse models were used to evaluate the anti-tumor effect of HTHSG under AMF (H = 1.3 kA/m, f = 255 kHz). The details are available in Supplementary Movie 2.

## Feasibility assessment of HTHSG in Beagle dog

A healthy adult female beagle dog weighing 7 kg was selected in this study. All procedures were performed under general anesthesia. The dog was intravenous anesthesia with sumianxin II (10–15 mg/kg), and endotracheal intubation. Continuous monitoring was performed by electrocardiography and oxygen saturation by reflectance oximetry using a sensor clipped to the ear. The perianal region was shaved, and the beagle dog was placed in the lateral position. The perineum and anal areas were then disinfected with 70% ethanol and iodine tincture and covered with a sterile drape.

After preparation, a Zebra$^{TM}$ guidewire is placed to a designated position in the colorectum of the beagle dog by colonoscopy (RAE-109, JeetMed, China). At the same time, HTHSG is placed between the gastric sac and esophageal sac of the Sengstaken-Blakemore Tube. Then the Sengstaken-Blakemore Tube loaded with HTHSG is placed along the guidewire into the designated position in the colorectum. After placement, we inflated the gastric sac and HTHSG was successfully fixed in the position.

Furthermore, a polylactic acid bracket was 3D printed to better maintain the structural and functional integrity of HTHSG. The HTHSG was placed in the groove in part 1 of the bracket, the part 2 was then twisted with the groove in part 1. This composition is then placed between the gastric sac and esophageal sac of the Sengstaken-Blakemore Tube and similarly placed in the colorectum as described above. After that, the beagle was exposed to AMF (H = 9.8 kA/m, f = 284 kHz) in the same way as the mice.

## Feasibility application of HTHSG in a human-sized Dummy

We conducted experiments in a dummy enema training model with similar size as human. HTHSG was placed in to the colorectum of a dummy and exposed to AMF (H = 114 kA/m, f = 20 kHz, HX120, HeXin, China) for 30 min. The dummy was placed into an AMF coil with 32 cm in diameter. To verify the safety of this treatment, we implanted HTHSG rectally in a rabbit at the same AMF and AMF coil conditions as a dummy. The rabbit was exposed to the AMF for 30 minutes on days 1, 3, and 5.

## Immune response analysis in vivo

To evaluate the immune response in vivo, spleens were collected from C57BL/6 mice after various treatments for flow cytometry. The spleens were cut into small pieces and digested with collagenase IV at 37 °C for 0.5 h. After gradient centrifugation, single-cell suspensions were collected for fluorescent antibody staining and subsequent flow cytometry. The single-cell suspensions from spleens were stained with anti-MHCII-BV510 and anti-CD11c-PE to analyze the DC percentage. At the same time, anti-CD3-BV510, anti-CD4-FITC, and anti-CD8-BV786 were used to examine the percentage of CD4$^+$ and CD8$^+$ T cells. Furthermore, anti-CD3-FITC, anti-CD4-BV786, anti-CD25-PE, and anti-Foxp3-APC antibodies were used to stain the single-cell suspensions to analyze regulatory T cells.

## CT imaging in vivo

CT imaging in vivo was carried out to evaluate the effect of hydrospongel on the surrounding tissue and organs. For mice, in vivo CT imaging was performed using a micro-CT (Bruker, SkyScan 1276). Imaging parameters were as follows: object to source, 91 mm; camera to source,158 mm; source voltage, 70 kV; source current, 200 µA; image pixel size, 20 µm; scaled image pixel size, 20 µm. CT images were analyzed using DataViewer. All mice were anesthetized with isoflurane gas and remained anesthetized throughout the image collection process.

For the beagle, in vivo CT imaging was performed using an X-ray CT (Supria, HITACHI, Japan). Imaging parameters were as follows: source voltage, 120 kV; source current, 150 mA; scaled image pixel size, 32 µm. The CT images were analyzed using DataViewer. The beagle was given intravenous anesthesia with sumianxin II (10–15 mg/kg) and remained anesthetized throughout the image collection process.

## Defecography

Defecography was performed before and after treatment to evaluate anal function. The beagle was kept in lateral recumbency for defecation imaging. Approximately 20 mL of barium paste (barium solution, water, and bran) was injected into the rectum. Three dynamic states (at rest, defecation, and evacuation) were photographed using digital radiography (ipet-400; IWA Biotech Co., Ltd, China).

## Immunohistochemistry and H&E staining

The expression of proteins in tumor tissues was validated by immunohistochemistry, and the biosafety of hydrospongel was assessed by H&E staining. Fixed tissues were embedded in paraffin and cut into 4 μm thick slices for immunohistochemistry staining. Paraffin sections were heated at 67 °C for 1 h. After routine deparaffinization and rehydration in xylene and an ethanol gradient, antigen retrieval was performed in a 121 °C high-pressure boiler with a 0.01 M sodium citrate buffer. Tissue sections were blocked with 5% BSA and incubated with specific primary antibodies (CK, KI67, MLH1, HER2, GPX4, Cleaved Caspase 3, and HSP70) overnight at 4 °C. Next, the tissue sections were incubated with corresponding secondary antibodies at room temperature for 1 h and stained with DAB. Finally, the tissue slices were counterstained with hematoxylin and colored with lithium carbonate. For H&E staining, after deparaffinization and rehydration, the tissue sections were first stained with hematoxylin for 2 min and then placed in an HCl/EtOH solution (v/v, 1%) for 3 s. Subsequently, the slices were stained again using eosin for 1 min and rinsed with tap water for 5 min. All stained sections were scanned using a slide scanner (Axioscan 7, ZEISS, Germany).

## Statistics & reproducibility

This study showed all data as means ± standard deviation (SD). The Statistical differences between groups were analyzed by one-way ANOVA with Tukey's post hoc test or two-tailed unpaired Student's t-test. The statistical analysis was performed using OriginLab Pro (Education edition, OriginLab, U.S.A.), and a $p$-value less than 0.05 (*), 0.01 (**), 0.001 (***) or 0.0001 (****) was reported. Each experiment in this study was designed to use the minimum number of animals required to obtain informative results. Our previous experience with the different tumor models provided guidance about the adequate number of animals. Tumor-bearing mice were randomized before treatment. No data were excluded from the analysis.

## Reporting summary

Further information on research design is available in the Nature Portfolio Reporting Summary linked to this article.

## Data availability

All data generated or analyzed during this study are provided in the paper and the Supplementary Information. No additional datasets requiring deposition in external repositories were generated. Source data are provided with this paper.

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

## Acknowledgements

This work was sponsored by the following Projects: National Key Research and Development Program of China (2022YFA1105300 to Z. W.; 2023YFB4705705 to J.Z.); LiaoNing Revitalization Talents Program (XLYC2203051 to T.G.); the National Natural Science Foundation of China (32370771 to J.Z.); Key Research Project of LiaoNing (2022JH1/10800002 to T.G.).

## Author contributions

Z.W., T.G. and J.Z. conceived and supervised the study. T.W., Y.T. and H.L. synthesized nanoparticles, hydrogels, and drug-loaded hydrogel microspheres and characterized their physicochemical properties. J.Z., T.L., X.Y. and C.Z. participated in the characterization of the hydrogel's magneto-thermal properties. T.L., Y.H. and C.Z. performed cell experiments and animal experiments. T.W. and T.L. analyzed the data and wrote the manuscript. Z.W., T.G. and J.Z. revised the manuscript.

## Competing interests

The authors declare no competing interests.
