## [Transparent Peer Review file · Nature Communications]

A Hollow-Tube-Like Hydrospengel for Multimodal Therapy of Advanced Colorectal Cancer

Corresponding Author: Professor Zhenning Wang

Version 0:

Reviewer comments:

Reviewer #1

(Remarks to the Author)

The authors have made a considerable effort in revising the paper and have added a substantial amount of additional data. However, a key issue remains—the lack of detailed information regarding the experiments.

For example:

(1) In terms of magnetic hyperthermia therapy, what were the strength and frequency of the alternating magnetic field used for different animal studies? Are they different for different sizes of animals? for example, for mice, beagles? Does it require different AMF for human uses?

(2) Does the temperature increase vary with different animal species?

(3) For all the animal studies, the dosages including the concentration of all the drugs and compositions need to be provided in details, as well as the electromagnetic induction time, strength, frequency.

Furthermore, I agree with Reviewer 2's comment that the combination of multiple modalities is too complex to be translatable to clinical applications, despite the various animal studies conducted. Significant challenges remain, including the production of reproducible materials (such as hydrogels, iron oxide particles, and microspheres) and the invasive nature of the implantation approach

Reviewer #2

(Remarks to the Author)

Thank you for your thorough response to my comments. I appreciate the significant improvements you've made to the manuscript, particularly the development of the 5-FU microsphere delivery system that extends release time to 48 hours and the addition of crucial control experiments. However, there are still two important issues that need to be addressed:

1) The microfluidic method used for producing the drug-loaded hydrogel microspheres requires better documentation - the website you reference (Fluidiclab, China) isn't accessible internationally, so you should either include a published citation for this method or provide a clear schematic diagram of the experimental setup.

2) The immunohistochemistry (IHC) images in the figures still have insufficient resolution, making it impossible to properly evaluate cellular details when zoomed in. High-quality IHC images are essential for peer reviewers to properly assess your histological findings.

Addressing these final points will strengthen the technical rigor and reproducibility of your interesting work.

Reviewer #3

(Remarks to the Author)

Reviewer #4

(Remarks to the Author)

This study introduces a hollow-tube-like hydrosponge (HTHSG) integrating magnetothermal therapy, chemotherapy, and chemodynamic therapy for advanced colorectal cancer. This study developed a multifunctional platform and validated its significant anti-tumor effect, particularly suitable for pre-surgical treatment option for advanced-stage CRC. It can be accepted in the present form.

Version 1:

Reviewer comments:

Reviewer #1

(Remarks to the Author)

The authors have made additional efforts to further revise the paper, and my questions have been satisfactorily addressed

Reviewer #2

(Remarks to the Author)

I want to thank the authors for addressing all reviewers' comments. The manuscript can be accepted.

Reviewer #3

(Remarks to the Author)

Response to reviewer's comments of manuscript "A Hollow-Tube-Like Hydrogel for Magnetothermal-Chemo-Chemodynamic Therapies in Advanced Colorectal Cancer Patients" (Research Article, No. NCOMMS-25-25089A).

We greatly appreciate the reviewers' thoughtful and constructive comments, which have been invaluable in guiding the revision and improvement of our manuscript. We have carefully reviewed all the comments and have addressed each point in detail in our point-by-point responses below, which are highlighted in blue text for clarity. Additionally, all revised or newly added content in the manuscript is marked in yellow.

Detailed point-by-point response to reviewers' comments.

Response to Reviewer 1:

The authors have made a considerable effort in revising the paper and have added a substantial amount of additional data. However, a key issue remains—the lack of detailed information regarding the experiments.

For example:

(1) In terms of magnetic hyperthermia therapy, what were the strength and frequency of the alternating magnetic field used for different animal studies? Are they different for different sizes of animals? for example, for mice, beagles? Does it require different AMF for human uses?

We thank the reviewer for highlighting the need for more detailed information regarding the experimental conditions of magnetic hyperthermia therapy (MHT). Following your suggestion, we have expanded our manuscript to clarify the strength and frequency of the alternating magnetic field (AMF) used in studies involving different animal models and in a human-sized dummy. We have also discussed how AMF parameters may differ depending on the size of the subject and addressed implications for potential clinical translation.

To evaluate the feasibility of HTHSG for human application, we conducted experiments using a dummy enema training model with dimensions comparable to those of a human. An AMF coil with a 32 cm diameter was constructed to accommodate the dummy, and HTHSG was placed in the colorectum before being exposed to the AMF. Infrared thermal imaging confirmed a temperature increase in HTHSG. To further assess safety, we implanted HTHSG rectally in a rabbit and subjected it to the same AMF and coil parameters as the dummy.

In our study, HTHSG was exposed to AMF across three models of varying size: mice, beagles, and the human-sized dummy. Each model required appropriately sized AMF coils. Specifically, as the coil diameter increased, achieving effective magnetothermal performance became more challenging under identical AMF conditions. Therefore, different equipment was used for each model to ensure HTHSG could reliably exhibit a magnetothermal effect. In addition, as demonstrated in the dummy experiments, larger models—such as humans—may require longer exposure times to reach the desired therapeutic temperature. The specific AMF parameters (strength and frequency) utilized for mice, beagle, and dummy experiments are summarized in Supplementary Table 5. Importantly, the product of frequency and field strength in all cases remained below the accepted biological safety limit of 5×10^9 A/m/s. All AMF settings used in our study met this safety criterion.

Corresponding revisions and additional details are now included in the manuscript as shown below. Thank you again for your constructive feedback, which has significantly improved the clarity and rigor of our study.

Results and discussion

Feasible application of HTHSG

To further evaluate the feasibility of HTHSG for human application, we conducted experiments using a dummy enema training model with dimensions comparable to those of a human. An AMF coil with a 32 cm diameter was constructed to accommodate the dummy, and HTHSG was placed in the colorectum before being exposed to the AMF (Fig. 8o). Infrared thermal imaging confirmed a temperature increase in HTHSG to approximately 42°C under these conditions (Fig. 8p). To further assess safety, we

implanted HTHSG rectally in a rabbit and subjected it to the same AMF and coil parameters as the dummy (Fig. 8q). The rabbit underwent 30-minute exposures on days 1, 3, and 5, with temperature changes monitored by infrared thermal imaging (Fig. 8r). Histological analysis (Masson trichrome staining) revealed no atrophy or collagen deposition in the anal sphincters (Fig. 8s), while H&E staining of major organs showed no signs of tissue damage (Supplementary Fig. 32), confirming the biocompatibility of HTHSG under these conditions.

In our study, HTHSG was exposed to AMF across three models of varying size: mice, beagles, and the human-sized dummy. Each model required appropriately sized AMF coils. Specifically, as the coil diameter increased, achieving effective magnetothermal performance became more challenging under identical AMF conditions^{62,63}. Therefore, different equipment was used for each model to ensure HTHSG could reliably exhibit a magnetothermal effect. In addition, as demonstrated in the dummy experiments, larger models—such as humans—may require longer exposure times to reach the desired therapeutic temperature. The specific AMF parameters (strength and frequency) utilized for mice, beagle, and dummy experiments are summarized in Supplementary Table 5. Importantly, the product of frequency and field strength in all cases remained below the accepted biological safety limit of 5×10^9 A/m/s⁶⁴. All AMF settings used in our study met this safety criterion.

In summary, our data demonstrate that the magnetothermal effect of HTHSG can be achieved across a range of model sizes by adapting AMF coil size and exposure time. The safety and biocompatibility of these conditions were further supported by histological analysis. For human application condition, the safety of treatment was verified in a rabbit. However, we acknowledge that before clinical application in humans, the long-term safety of this approach will require further investigation.

Fig. 8 | Feasible application of HTHSG. **o**, Schematic of HTHSG hydrospongel implantation and treatment in a dummy enema training model. **p**, Thermal images of dummy enema training model during AMF exposure. **q**, Schematic of HTHSG hydrospongel implantation and treatment in a rabbit. **r**, Thermal images of rabbit during AMF exposure. **s**, Representative Masson trichrome staining images of rabbit sphincter. Scale bar: 0.5 mm.

Supplementary Fig. 32 | Representative H&E staining images of major organs including heart, liver, spleen, lung, and kidney of rabbit. Scale bar: 100 μm .

Supplementary Table 5. AMF application conditions for different species

Species	Coil Diameter	Frequency	Field Strength	Time for HTHSG to reach 42°C	Exposure Period
Mice	5 cm	255 KHz	1.3 KA/m	Around 2 min	10 min
Beagle	16 cm	284 KHz	9.8 KA/m	Around 3-4 min	10 min
Human (Dummy)	32 cm	20 KHz	114.0 KA/m	Around 15 min	30 min

Reference

- 62 Nian, S.-C., Tsai, S.-W., Huang, M.-S., Huang, R.-C. & Chen, C.-H. Key parameters and optimal design of a single-layered induction coil for external rapid mold surface heating. *International Communications in Heat and Mass Transfer* **57**, 109-117, (2014).
- 63 Todaka, T. & Enokizono, M. Optimal design method with the boundary element for high-frequency quenching coil. *IEEE Transactions on Magnetics* **32**, 1262-1265, (1996).
- 64 Çelik, Ö., Can, M. M. & Firat, T. Size dependent heating ability of CoFe_2O_4 nanoparticles in AC magnetic field for magnetic nanofluid hyperthermia. *Journal of Nanoparticle Research* **16**, 2321, (2014).

(2) Does the temperature increase vary with different animal species?

We thank the reviewer for this insightful question. As discussed above, we assessed the potential application of HTHSG in humans using a human-sized dummy model, in addition to experiments in mice and beagles. In our study, HTHSG was exposed to AMF in all three models—mice, beagle, and dummy—with AMF coils of different sizes tailored to each model.

We observed that the size of the AMF coil significantly influences the magnetothermal performance of HTHSG. Specifically, larger AMF coils (required for larger species or

models) make it more challenging to achieve the same degree of temperature elevation under identical field conditions. Consequently, different equipment and AMF settings were necessary to ensure the effective magnetothermal response of HTHSG in each model.

Moreover, the time required for HTHSG to reach the target temperature (~42°C) increased with model size. In mice and beagle models, the desired temperature was reached in less than 5 minutes, while in the dummy model (human application), it took approximately 15 minutes. These results are summarized in Supplementary Table 5 and have been added to the revised manuscript for clarity.

All AMF parameters used in our experiments complied with the established biological safety limit (the product of frequency and field strength did not exceed 5×10^9 A/m/s), and the safety and biocompatibility of HTHSG were confirmed by H&E staining of major organs. For the human application condition, safety was further verified in the rabbit model. However, before translation to clinical practice, further investigation into long-term safety is warranted.

These clarifications and data have been incorporated into the revised manuscript as indicated.

Supplementary Table 5. AMF application conditions for different species

Species	Coil Diameter	Frequency	Field Strength	Time for HTHSG to reach 42°C	Exposure Period
Mice	5 cm	255 KHz	1.3 KA/m	Around 2 min	10 min
Beagle	16 cm	284 KHz	9.8 KA/m	Around 3-4 min	10 min
Human (Dummy)	32 cm	20 KHz	114.0 KA/m	Around 15 min	30 min

(3) For all the animal studies, the dosages including the concentration of all the drugs and compositions need to be provided in details, as well as the electromagnetic induction time, strength, frequency.

We thank the reviewer for this excellent suggestion. In response, we have now provided detailed information on the dosages and concentrations of all drugs and components used in the animal studies, as well as the electromagnetic induction time, field strength, and frequency. This information has been comprehensively summarized in Supplementary Table 5 and Supplementary Table 7. The revised manuscript now includes the following:

Materials and Methods

Drug Loading and Release

A 5-FU aqueous solution was added dropwise to lyophilized HTHSG to prepare the drug-loaded hydrospengel, and its 5-FU release profile was subsequently evaluated. The 5-FU microsphere-loaded HTHSG was fabricated using the same method as described previously for HTHSG, with the exception that 1.16 g of 5-FU-loaded hydrogel microspheres were added to 5 mL of deionized water along with Fe₃O₄@PDA nanoparticles to achieve drug loading. Further experimental details are provided in the Supplementary Information (SI) Experimental Procedures. Typically, 5 mL of the hydrogel precursor solution yields approximately 250 mg of hydrospengel. The drugs and compositions used in all animal studies and experimental groups are summarized in Supplementary Table 7.

Supplementary Table 7. Detailed information on drug and compositions used in all animal studies

Animal studies	Groups	Statements	Hydrospongel	Fe ₃ O ₄ @PDA NPs	5-FU	AMF	
Mice (CDX/PDX/ Orthotopic Models)	Group 1	PBS+AMF group	-	-	-	Yes	
	Group 2	CP10+AMF group	25 mg rectally	-	-	Yes	
	Group 3	CFP20 group	25 mg rectally	About 10 mg	-	-	
	Group 4	CP20+5-FU MSs +AMF group	25 mg rectally	-	About 1mg 5-FU in 25 mg hydrospongel	Yes	
	Group 5	5-FU+AMF group	-	-	Total 1 mg 5-FU intraperitoneally injected (0.5 mg per time)	Yes	
	Group 6	CFP20+AMF group	25 mg rectally	25 mg rectally	About 10 mg	-	Yes
	Group 7	CFP20+5-FU MSs +AMF group	25 mg rectally	25 mg rectally	About 10 mg	About 1mg 5-FU in 25 mg hydrospongel	Yes
Beagle	-	CFP20+5-FU MSs (HTHSG) +AMF	150 mg rectally	About 60 mg	About 6mg 5-FU in 150 mg hydrospongel	Yes	

Supplementary Table 5. AMF application conditions for different species

Species	Coil Diameter	Frequency	Field Strength	Time for HTHSG to reach 42°C	Exposure Period
Mice	5 cm	255 KHz	1.3 KA/m	Around 2 min	10 min
Beagle	16 cm	284 KHz	9.8 KA/m	Around 3-4 min	10 min
Human (Dummy)	32 cm	20 KHz	114.0 KA/m	Around 15 min	30 min

Furthermore, I agree with Reviewer 2's comment that the combination of multiple modalities is too complex to be translatable to clinical applications, despite the various animal studies conducted. Significant challenges remain, including the production of reproducible materials (such as hydrogels, iron oxide particles, and microspheres) and the invasive nature of the implantation approach

We thank the reviewer for this insightful comment. In response, we conducted experiments using a human-sized dummy model to further assess the feasibility of HTHSG for clinical application. Our aim is to develop a novel preoperative strategy for colorectal cancer patients by integrating synergistic magnetothermal, chemo-, and chemodynamic therapies, thereby improving the likelihood of radical tumor resection and anal preservation. We acknowledge that further studies are necessary to confirm the efficacy and biosafety of HTHSG in humans. Clinical trials will be essential to establish its anti-tumor effectiveness and safety profile. Moreover, widespread clinical adoption of HTHSG will require financial investment in AMF-generating equipment and close interdisciplinary collaboration to optimize and streamline colonoscopic implantation procedures.

Response to Reviewer 2:

Thank you for your thorough response to my comments. I appreciate the significant improvements you've made to the manuscript, particularly the development of the 5-FU microsphere delivery system that extends release time to 48 hours and the addition of crucial control experiments. However, there are still two important issues that need to be addressed.

1) The microfluidic method used for producing the drug-loaded hydrogel microspheres requires better documentation - the website you reference (Fluidiclab, China) isn't accessible internationally, so you should either include a published citation for this method or provide a clear schematic diagram of the experimental setup

We thank the reviewer for this helpful suggestion. In this study, drug-loaded hydrogel microspheres were prepared using a microfluidic chip from FluidicLab (<https://en.fluidiclab.com>, China). In response to the reviewer's recommendation, we have now included both a digital photograph and a schematic diagram of the experimental setup in Supplementary Figure 18 to more clearly illustrate the preparation process of the drug-loaded hydrogel microspheres. In addition, we have expanded the description of our experimental methodology to provide further details on the devices and reagents used. The revised manuscript is shown as below.

Results

To prolong the release of 5-FU, a small hydrophilic chemotherapeutic drug, we fabricated drug-loaded hydrogel microspheres using microfluidic technology (Fig. 4h and Supplementary Fig. 18).

Supplementary Fig. 18 | a, The digital photograph of the experimental setup for the preparation of drug-loaded hydrogel microspheres. **b**, The schematic diagram of the preparation process of drug-loaded hydrogel microspheres.

Supplementary Experimental Procedures

Preparation of drug-loaded hydrogel microspheres

The drug-loaded hydrogel microspheres were prepared using a template of water-in-oil (W/O) single emulsion and generated in the polydimethylsiloxane microfluidic device with flow-focusing junction (PDMS-FF-100, Fluidiclab, China). Briefly, 5-FU (10 mg/mL) was first dissolved into a 10-fold PBS solution (pH = 7.4) under ultrasound, and gelatin (4% w/v) was then dissolved therein at 37°C to serve as the dispersed phase. In addition, the fluorinated oil (Drop-Surf droplet generation oil, FluidicLab, China)

was used as the continuous phase and all phases were injected into the microchannels at constant flow rates (0.1 mL/min) controlled by syringe pumps (LSP01-1Y, Rongbai Pump, China). The fabrication process was observed using a microscope (MMJ31, Microtomo, China), and the generated 5-FU-loaded gelatin emulsion droplets were collected with dialdehyde cellulose (DAC) aqueous solution (2% w/v) containing 5-FU (10 mg/mL) in a petri dish and immersed for 1.5 hours to allow cross-linking. The resulting hydrogel microspheres were subjected to an emulsion breaker (Drop-Surf demulsifier, FluidicLab, China) for 5 minutes, followed by washing with deionized water and collection by centrifugation a centrifuge (TD5, Yingtai Instrument, China) at 600 rpm for 5 minutes for subsequent assays.

2) The immunohistochemistry (IHC) images in the figures still have insufficient resolution, making it impossible to properly evaluate cellular details when zoomed in. High-quality IHC images are essential for peer reviewers to properly assess your histological findings.

Addressing these final points will strengthen the technical rigor and reproducibility of your interesting work.

We sincerely appreciate the reviewer's comments. In response to this suggestion, we have provided high-resolution IHC images and have uploaded each figure (Figures 1-8) as individual high-quality files to the submission system, in addition to including these figures in the combined manuscript document.

Response to Reviewer 3:

We would like to express our sincere gratitude to the reviewer for the time and effort devoted to evaluating our manuscript. We are committed to revising our manuscript according to the suggestions from all reviewers, as these recommendations are highly valuable for enhancing the quality of our research.

Response to Reviewer 4:

This study introduces a hollow-tube-like hydrospongel (HTHSG) integrating

magnetothermal therapy, chemotherapy, and chemodynamic therapy for advanced colorectal cancer. This study developed a multifunctional platform and validated its significant anti-tumor effect, particularly suitable for pre-surgical treatment option for advanced-stage CRC. It can be accepted in the present form.

We sincerely appreciate the reviewers' constructive comments throughout the review process and the recognition of our revisions. This valuable feedback has been instrumental in improving the content, quality, and overall integrity of our manuscript. Thank you for your time and effort in reviewing our work.

Yours sincerely,

Dr. Zhenning Wang, MD, Ph.D.

Department of Surgical Oncology and General Surgery, The First Hospital of China Medical University

Key Laboratory of Precision Diagnosis and Treatment of Gastrointestinal Tumors, Ministry of Education, China Medical University

155 North Nanjing Street

Shenyang, China, 110001

Email: znwang@cmu.edu.cn